# A rapid cell-free expression and screening platform for antibody discovery

Andrew C. Hunt ®[1,2], Bastian Vögeli ®[1,2], Ahmed O. Hassan[3], Laura Guerrero[1,2], Weston Kightlinger[1,2], Danielle J. Yoesep[1,2], Antje Krüger ®[1,2], Madison DeWinter ®[1,2,4], Michael S. Diamond ®[3,5,6,7], Ashty S. Karim ®[1,2] & Michael C. Jewett[1,2,8,9,10] ✉

Antibody discovery is bottlenecked by the individual expression and evaluation of antigen-specific hits. Here, we address this bottleneck by developing a workflow combining cell-free DNA template generation, cell-free protein synthesis, and binding measurements of antibody fragments in a process that takes hours rather than weeks. We apply this workflow to evaluate 135 previously published antibodies targeting the severe acute respiratory syndrome coronavirus 2 (SARS-CoV-2), including all 8 antibodies previously granted emergency use authorization for coronavirus disease 2019 (COVID-19), and demonstrate identification of the most potent antibodies. We also evaluate 119 anti-SARS-CoV-2 antibodies from a mouse immunized with the SARS-CoV-2 spike protein and identify neutralizing antibody candidates, including the antibody SC2-3, which binds the SARS-CoV-2 spike protein of all tested variants of concern. We expect that our cell-free workflow will accelerate the discovery and characterization of antibodies for future pandemics and for research, diagnostic, and therapeutic applications more broadly.

Antibodies are widely used in diagnostics and as drugs. They are the critical component in immunoassays enabling rapid diagnostics[1] and constitute one of the fastest-growing classes of therapeutics with 18 to 30% of new FDA-approved drugs in the last two years being antibodies[2–5]. Antibodies have also recently garnered attention as potential countermeasures for emerging pathogens, having been used as both prophylaxis and therapy against infection with severe acute respiratory syndrome coronavirus 2 (SARS-CoV-2) virus and coronavirus disease 2019 (COVID-19)[6].

Contemporary workflows for antibody discovery commonly utilize synthetic selections or the isolation of single B cell clones from convalescent patients or animals to go from >$10^8$ sequences to a pool of $10^3$ to $10^4$ candidates targeting the desired antigen. However, once this pool of candidates is generated, state-of-the-art workflows rely on labor-intensive procedures (e.g., plasmid-based cloning, transfection, cell-based protein expression, protein purification, binding assessment through enzyme-linked immunosorbent assays (ELISAs), etc.) to evaluate and identify the best antibody candidates[7,8]. These procedures take weeks to months and represent a major speed and throughput bottleneck in antibody discovery.

The effort to identify antibodies against emerging threats like SARS-CoV-2 during the coronavirus disease 2019 (COVID-19) pandemic

[1]Department of Chemical and Biological Engineering, Northwestern University, Evanston, IL 60208, USA. [2]Center for Synthetic Biology, Northwestern University, Evanston, IL 60208, USA. [3]Department of Medicine, Washington University School of Medicine, St. Louis, MO 63110, USA. [4]Medical Scientist Training Program, Northwestern University Feinberg School of Medicine, Chicago, IL 60611, USA. [5]Department of Molecular Microbiology, Washington University School of Medicine, St. Louis, MO 63110, USA. [6]Department of Pathology & Immunology, Washington University School of Medicine, St. Louis, MO 63110, USA. [7]Andrew M. and Jane M. Bursky Center for Human Immunology and Immunotherapy Programs, Washington University School of Medicine, St. Louis, MO 63110, USA. [8]Chemistry of Life Processes Institute, Northwestern University, Evanston, IL 60208, USA. [9]Robert H. Lurie Comprehensive Cancer Center, Northwestern University, Chicago, IL 60611, USA. [10]Department of Bioengineering, Stanford University, Stanford, CA 94305, USA. ✉e-mail: mjewett@stanford.edu

has highlighted the importance of (i) rapid and high-throughput antibody discovery platforms and (ii) identifying high-affinity antibodies targeting conserved[9,10] or non-overlapping epitopes[11,12] to resist viral escape and increase the ability to neutralize viral variants[6,13]; both of which require intensive screening campaigns. A further challenge is that existing antibody discovery processes frequently have low efficiency, with few of the screened candidates having potent neutralizing activity, as has been the case for SARS-CoV-2 (Supplementary Table 1). Taken together, the limitations in existing antibody discovery processes suggest an urgent need for faster and higher throughput screens.

Cell-free protein synthesis (CFPS)[14,15], the manufacture of proteins without living cells using crude extracts or purified components, is an attractive tool to overcome these limitations. A variety of CFPS systems for antibody expression have been developed[16–22]; however, few of these studies have focused on the functional screening of antibodies, and most methods rely on techniques that are not suitable for high-throughput screening like the use of purified plasmids or labor-intensive ELISAs[16,18–20,23].

Here, we describe a CFPS-based integrated pipeline for antibody expression and evaluation to address screening limitations in current antibody discovery pipelines. The workflow leverages four key developments (Fig. 1a): (i) DNA assembly and amplification methods that do not require living cells, (ii) CFPS systems that work directly from linear DNA templates and generate disulfide-bonded antibody molecules, (iii) an Amplified Luminescent Proximity Homogeneous Linked Immunosorbent Assay (AlphaLISA) that enables rapid protein-protein interaction (PPI) characterization without protein purification[24], and (iv)

acoustic liquid handling that enables a highly parallel and miniaturized workflow. This integrated workflow enables a single researcher to express and profile the antigen-specific binding of hundreds of antibodies in less than 24 h. As a model, we apply our workflow to profile a diverse set of 135 previously published antibodies targeting the SARS-CoV-2 spike glycoprotein, and show that our workflow identifies all 8 neutralizing antibodies that had been granted emergency use authorization (EUA) by the United States Food and Drug Administration for the treatment of COVID-19. In addition, we screen 119 antibodies derived from mice immunized against the SARS-CoV-2 spike glycoprotein and identify several candidate neutralizing antibodies.

## Results

### Development of a cell-free DNA assembly and amplification workflow

We first implemented a method for cell-free DNA assembly and amplification by adapting and optimizing recently reported protocols for rapid construction of DNA templates for CFPS[16,18,23,25]. The method consists of a Gibson assembly step, followed by PCR amplification of the linear expression template (LET) using the unpurified Gibson assembly product as a template. The key idea was to create a versatile approach for construction of DNA templates without the requirement of cell culture, allowing for DNA assembly and amplification in less than 3 h entirely in 384-well plates.

To validate the method, we applied it to the assembly and amplification of a LET for sfGFP. We only observed sfGFP expression in the presence of properly assembled DNA template (Supplementary

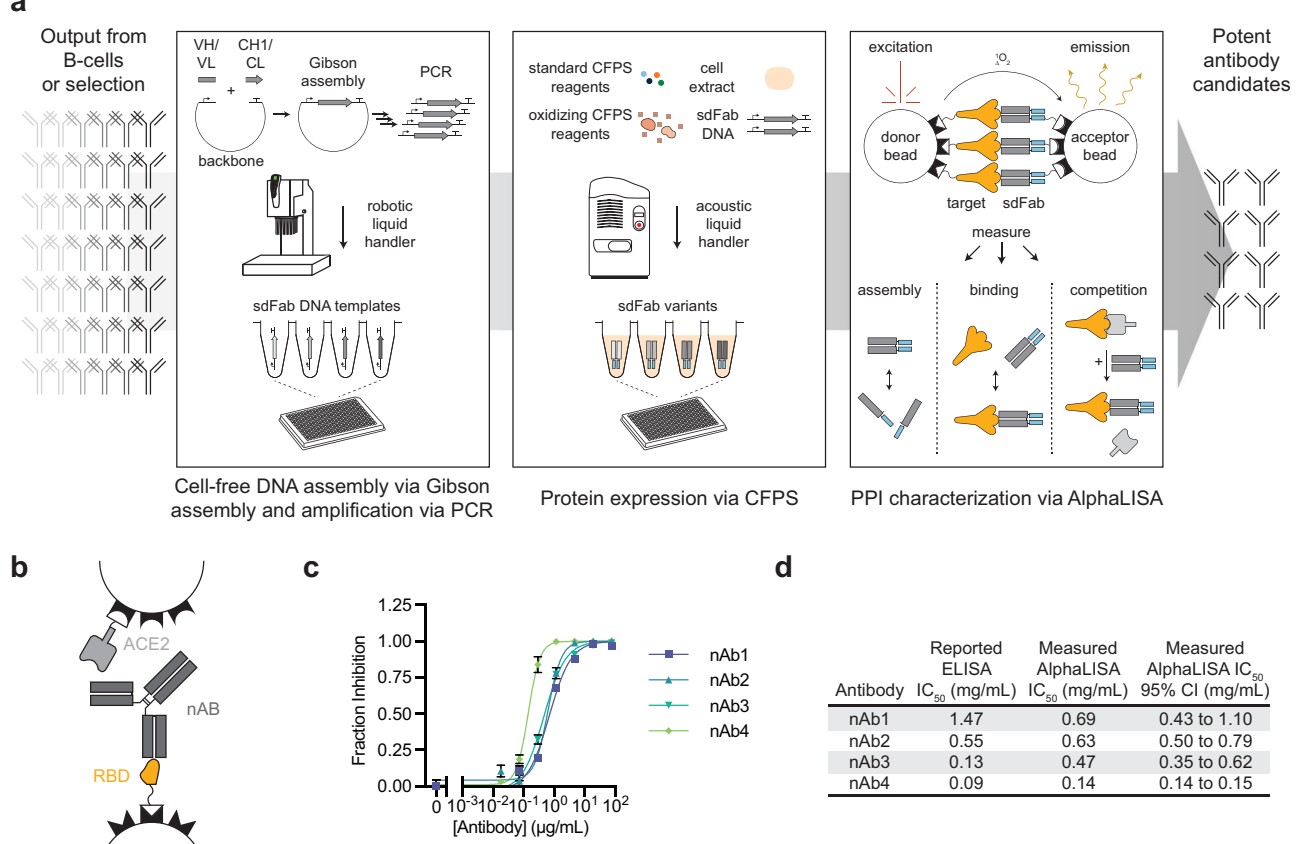

**Fig. 1 | A rapid cell-free antibody screening workflow. a** Schematic of the steps involved in the cell-free antibody screening workflow. **b** Diagram of the AlphaLISA screen for neutralizing antibodies via competition with angiotensin-converting enzyme 2 (ACE2) for the SARS-CoV-2 receptor binding domain (RBD). **c** Evaluation of commercial neutralizing antibodies (nAbs) via an AlphaLISA ACE2 competition screen ($n = 3$ independent replicates ± SEM). **d** Comparison of the reported and measured potencies of commercial neutralizing antibodies. Source data are provided in the Source data file.

| Antibody | Reported ELISA IC$_{50}$ (mg/mL) | Measured AlphaLISA IC$_{50}$ (mg/mL) | Measured AlphaLISA IC$_{50}$ 95% CI (mg/mL) |
|---|---|---|---|
| nAb1 | 1.47 | 0.69 | 0.43 to 1.10 |
| nAb2 | 0.55 | 0.63 | 0.50 to 0.79 |
| nAb3 | 0.13 | 0.47 | 0.35 to 0.62 |
| nAb4 | 0.09 | 0.14 | 0.14 to 0.15 |

Fig. 1a–c). To assemble antibody DNA templates, we purchased synthetic, double-stranded linear DNA coding for the desired variable heavy (VH) and variable light (VL) chain sequences. These DNAs were assembled with DNA coding for the appropriate heavy chain constant (CH1) or light chain constant (CL) antigen-binding fragment (Fab) domains in addition to a separate piece of DNA coding for the backbone of the pJL1 vector[26]. These sequences were subsequently amplified by PCR to generate LETs (Supplementary Fig. 1d–f). In addition to speed, this workflow also affords flexibility, where a single variable fragment can be assembled into different antibody formats (e.g., full-length heterotetrameric IgG, Fab, synthetically dimerized Fab (sdFab), etc.) containing different purification or immobilization tags by including different antibody constant regions in the assembly reaction.

## Development of a crude extract-based CFPS system for antibody fragment expression

We next demonstrated rapid antibody expression in a crude *E. coli*-based CFPS system. We developed a high-yielding ($1391 \pm 32\,\mu g/mL$ sfGFP, Supplementary Fig. 1c) crude *E. coli* lysate-based CFPS system from the Origami™ B(DE3) strain, which contains mutations in the *E. coli* reductase genes *trxB* and *gor* to enable the formation of disulfide bonds in the cytoplasm[27]. By pretreating the extract with the reductase-inhibitor iodoacetamide (IAM) to further stabilize the redox environment[28–30] and supplementing the reaction with purified *E. coli* disulfide bond isomerase DsbC and prolyl isomerase FkpA[19,31,32], we successfully expressed and assembled full-length trastuzumab (Supplementary Fig. 2a), a model anti-HER2 antibody[33], from linear DNA templates. Under the tested conditions, we observed additional bands corresponding to incomplete assembly of the IgG consistent with other reports that efficient assembly of full-length antibodies in CFPS requires further optimization (e.g., temperature, DNA template ratio, and DNA template expression timing)[16–20,31]. Consistent assembly under the same experimental conditions is important for screening since individual conditions cannot be optimized for hundreds or thousands of antibodies. We thus opted to use the synthetically dimerized antigen-binding fragment (sdFab, also called ecobodies[16,18] or zipbodies[18]) format and, like Ojima-Kato et al.[16–18], found that the assembly of sdFabs were more uniform than their corresponding Fabs in CFPS for a panel of antibodies (Supplementary Fig 2b, c). After establishing a protocol to express sdFabs, we used acoustic liquid handling to assemble 2 μL CFPS reactions in 384-well plates (Fig. 1a).

## Integration of the AlphaLISA PPI assay to evaluate antibody fragment binding and assembly

Following DNA assembly and CFPS, antigen-specific binding was evaluated. To characterize PPIs of the expressed sdFab antibody candidates, we developed AlphaLISA methods to evaluate binding directly from CFPS reactions. AlphaLISA is an in-solution and wash-free assay that is designed for high-throughput screening and is compatible with crude cell-lysates[24]. In AlphaLISA, capture chemistries are used to immobilize the proteins of interest on donor and acceptor beads, which generate a chemiluminescent signal when in proximity to one another and excited by a 680 nm laser[34]. We developed AlphaLISA methods to measure direct binding to an antigen, competition between two binders for a given epitope, bridging of an antigen with another binder targeting an orthogonal epitope, and assembly of protein complexes (Supplementary Fig. 3).

We first validated that AlphaLISA is tolerant of crude CFPS reactions. We observed that CFPS does not interfere with the measurement chemistry (Supplementary Fig. 4a), but that certain reaction components can disrupt protein immobilization to the bead, which can be circumvented with the appropriate choice of immobilization chemistry (Supplementary Fig 4b, c). For example, Ni-Chelate beads were not tolerant of the high salt concentrations and high concentration of

histidine present in CFPS, due to charge screening and Ni chelation, respectively, hindering immobilization of the polyhistidine-tagged proteins.

To validate the ability of AlphaLISA to profile neutralizing antibodies, we tested the capacity of four different commercial antibodies to compete with the SARS-CoV-2 target human receptor Angiotensin-Converting Enzyme 2 (ACE2) for binding of the SARS-CoV-2 Receptor Binding Domain (RBD). Our determined rank order of $IC_{50}$ values aligned well with the reported ELISA $IC_{50}$s (Fig. 1b–d).

Furthermore, we utilized AlphaLISA to develop a sdFab assembly screen to monitor antibody fragment expression and assembly in CFPS, a step that traditionally requires SDS-PAGE. The measurement immobilizes the heavy and light chains of the sdFab to the AlphaLISA beads, resulting in signal when the two chains are assembled (Supplementary Fig. 5a). The AlphaLISA assembly assay generally shows consistent prediction of sdFab assembly with SDS-PAGE on a panel of sdFabs and can thus be used as a heuristic to identify when sdFab expression or assembly fails (Supplementary Fig. 5b). In this panel, the sdFab REGN10933 yielded lower assembly signal than the other tested antibodies despite a strong band present by SDS-PAGE (Supplementary Fig. 2c). This could be a result of misfolding of the light chain constant domain, leading to reduced binding of the anti-light chain antibody to the REGN10933 light chain and thus lower assembly signal. However, deeper structural analysis of CFPS-derived REGN10933 would be required to understand this result further (Supplementary Fig. 3). Accordingly, we utilized the assembly assay as a qualitative positive control to confirm that antibodies expressed and assembled in the CFPS reaction and did not attempt to use the assay to quantify assembled antibody yields.

## Evaluation of a large set of previously published antibodies

Using the developed workflow, we next evaluated a set of 115 SARS-CoV-2 targeted antibodies that were selected based on the availability of sequence, structural, binding, and neutralization data, with 84 being drawn from Brouwer et al.[35] and the remaining 31 coming from diverse sources[36–48]. The antibodies span four orders of magnitude in neutralization potency and target a variety of domains and epitopes (Supplementary Data 1 and 2). Each antibody sequence was evaluated in the sdFab format using AlphaLISA to measure binding to the SARS-CoV-2 hexaproline stabilized pre-fusion spike glycoprotein (S6P)[49], binding to the SARS-CoV-2 RBD, competition with ACE2 for RBD binding, and assembly of the sdFab heavy and light chains in CFPS (Fig. 2, Supplementary Fig. 6). Measurements were considered hits if they were >3 standard deviations above the background and exhibited a *p*-value of <0.05 using a two-sided student's t-test corrected for multiple testing using the Benjamini and Hochberg False Discovery Rate procedure (FDR)[50]. We used these assays as a rapid screen for S6P binding and ACE2 competition at a single unknown concentration of CFPS-derived antibody.

To determine the robustness of the workflow, antibody fragments were expressed and evaluated in triplicate. We observed that independent replicates were consistent with one another and exhibited average coefficients of variation (standard deviation divided by the mean) in the range of 0.15–0.22 (Supplementary Fig. 7), indicating that the liquid handling and measurement workflow is reproducible.

Within the diverse set of 31 SARS-CoV-2 targeted antibodies, we observed assembly for 31 out of 31 tested antibody fragments, S6P binding for 24 out of 31 antibody fragments reported to bind the S6P, RBD binding for 19 out of the 30 antibody fragments reported to bind the RBD, and ACE2 competition for 11 out of 26 antibody fragments reported to compete with ACE2 (Fig. 2a). The sdFab 4A8, an N-terminal domain targeted antibody[46], only showed strong interaction with the S6P and the sdFab CR3022, whose target epitope is occluded in the pre-fusion S6P conformation[43,51], showed binding to the RBD, but weak binding to the S6P.

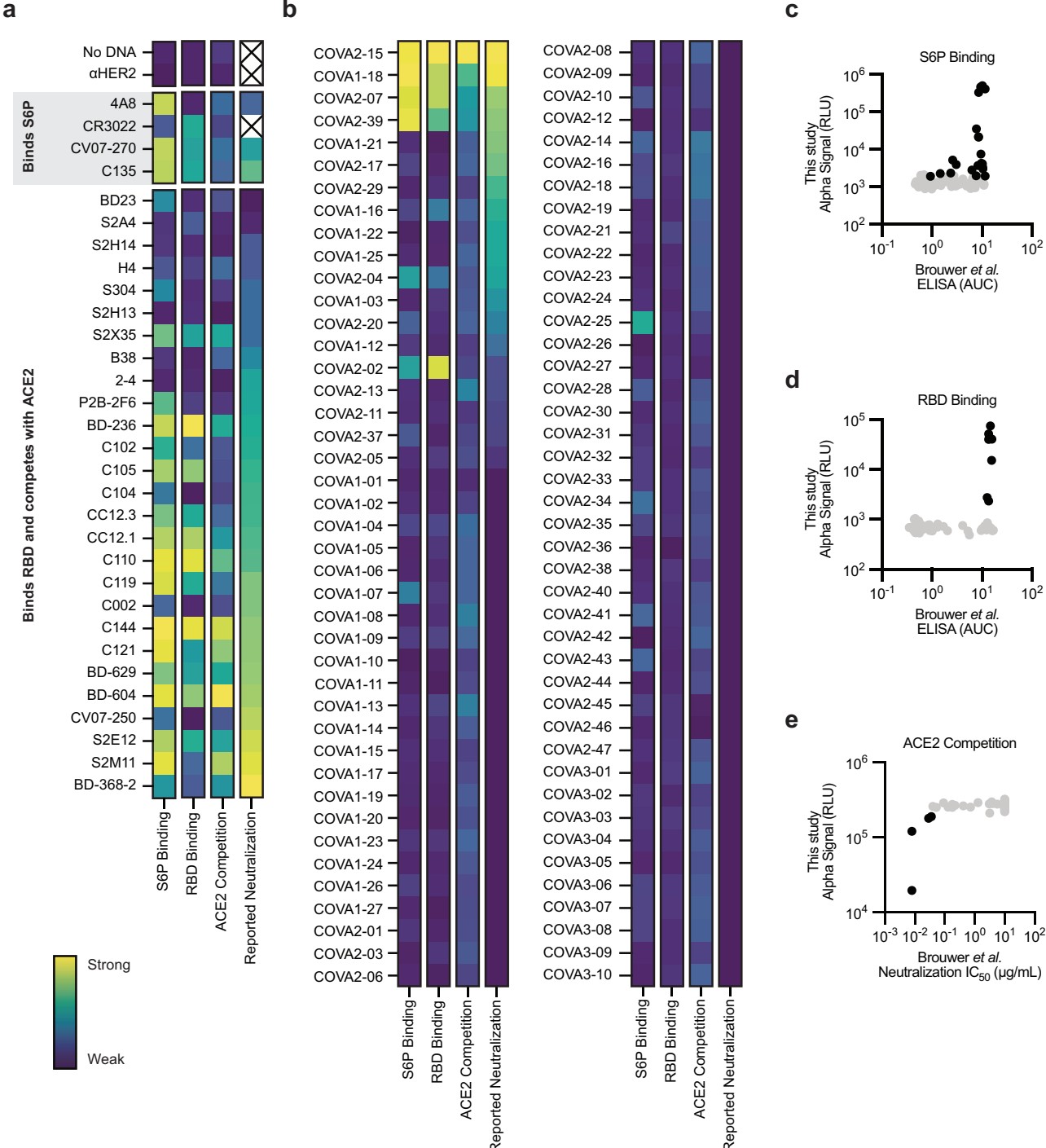

**Fig. 2 | Performance of the cell-free antibody fragment screening workflow evaluated on SARS-CoV-2 neutralizing antibodies. a, b** Heatmap of the binding of previously published antibodies measured using AlphaLISA to detect S6P binding (log₁₀ scaled), RBD binding (log₁₀ scaled), and ACE2 competition (linearly scaled). The lowest reported neutralization IC₅₀ value is also plotted for comparison (log₁₀ scaled) and an X indicates no relevant data available (Supplementary Data 1). **a** Heatmap of the binding of sdFabs derived from diverse sources. **b** Heatmap of the binding of sdFabs in the Brouwer et al. data set. **c–e** Parity plots comparing AlphaLISA measurements of the Brouwer et al. sdFabs vs. the previously

reported Brouwer et al. data. **c** S6P binding AlphaLISA vs. S6P binding ELISA. **d** RBD binding AlphaLISA vs RBD binding ELISA. **e** ACE2 competition AlphaLISA vs pseudovirus neutralization IC₅₀. **c–e** Black data points represent data that are statistically significant (two-sided t-test FDR corrected $p < 0.05$) from background and have an average value >3 standard deviations away from background. Grey data were considered not significantly different from background. **a–e** AlphaLISA data are the mean of 3 independent replicates ($n = 3$) derived from independent CFPS reactions. Antigen concentrations for AlphaLISA experiments listed in Supplementary Table 2. Source data are provided in the Source data file.

For the set of 84 antibodies from Brouwer et al., we observed assembly of 80 out of 84 antibody fragments and binding to the S6P and RBD for many of the antibody fragments that showed strong binding via ELISA (Fig. 2b–d). We also compared ACE2 competition

against the reported neutralization IC₅₀ value since it has been reported that more than 90% of neutralizing antibodies block the RBD and ACE2 interaction[38,52] and similar competition assays have been reported to correlate with neutralization potency[38,53] (Fig. 2e). We

observed ACE2 competition, as well as strong S6P and RBD binding, for 4 out of 5 antibody fragments that were determined to compete with ACE2, which also represent the four most potent neutralizing antibodies in the Brouwer et al. data set.

While there is only a weak correlation between our AlphaLISA data and the corresponding Brouwer et al. S6P binding ELISA data, the RBD binding ELISA data, and the pseudovirus neutralization data (Two-sided Pearson correlation coefficients with 95% confidence interval and $p$ value of $r = 0.35$ (0.15 to 0.53, $p = 0.01$), $r = 0.41$ (0.22 to 0.58, $p < 0.0001$), and $r = 0.40$ (0.20 to 0.56, $p = 0.002$) for Figs. 2c, 2d, and 2e, respectively), the screening conditions used consistently identified the strongest binders and most potent neutralizing antibodies. Collectively, these data show that our workflow can be used to express and evaluate human antibody fragments as a filter to select potential candidates for further development.

## Expression and evaluation of high interest SARS-CoV-2 antibodies

We next expressed and evaluated all 8 antibodies previously granted emergency use authorization (EUA) by the US Food and Drug Administration for prophylaxis or treatment of COVID-19 (LY-CoV555, CB6, REGN10933, REGN10987, S309, AZD8895, AZD1061, and LY-CoV1404)[9,54–59] as well as 11 other antibodies reported to broadly neutralize SARS-CoV-2 and related variants (SARS2-38, S2P6, S2X259, DH1047, C118, C022, S2K14, S2H97, A23-58.1, B1-182.1, and 54042-4)[10,60–67]. For all 8 EUA antibody fragments, we observed S6P binding, RBD binding, ACE2 competition, and assembly (Fig. 3a, Supplementary Fig. 8). These results are consistent with the literature on these antibodies, with the exception of S309, where we observed weak ACE2

competition despite previous structural and biophysical data suggesting that this antibody does not compete with ACE2[9]. To explore this result further, we performed a dose-dependent ACE2 competition and an RBD bridging experiment with ACE2 and S309 (Supplementary Figs. 3 and 9). Unexpectedly, we observed that ACE2 inhibits the S309 interaction with the RBD, but that S309 can also bind to the RBD at the same time as ACE2.

For the set of broadly neutralizing antibodies (bnAbs), we observed binding and competition for 10 of 11 antibody fragments tested, with the results largely consistent with published literature[10,60–67]. Notably, we observed that S2H97 exhibited ACE2 competition, which like S309, is reported to bind an epitope adjacent to the receptor binding motif and not compete with ACE2 for binding[10]. The antibody 54042-4 exhibited weak binding to the RBD but showed low assembly signal, indicative of poor expression or assembly (Supplementary Fig. 8). S2P6 is the only tested bnAb that does not target the RBD[61], which is borne out in our data (Fig. 3a). Based on these initial screens, our workflow would have identified 8 of 8 antibodies granted an EUA for the treatment of COVID-19 as well as 10 of 11 of the tested previously identified broadly neutralizing SARS-CoV-2 antibodies.

We further profiled the binding of this set of high interest antibodies against SARS-CoV-2 variants of concern (VOC) including Alpha (B.1.1.7), Beta (B.1.351), Gamma (P.1), Delta (B.1.617.2), and Omicron (BA.1, BA.2, BA.2.12.1, and BA.4/5) as well as several other human coronaviruses including SARS-CoV, MERS-CoV, HCoV-HKU1, HCoV-OC43, HCoV-NL63, and HCoV-229E (Fig. 3b). Our data for the EUA Abs against different VOC are consistent with previously reported reductions in neutralization potency (Supplementary Fig. S10)[68]. For these

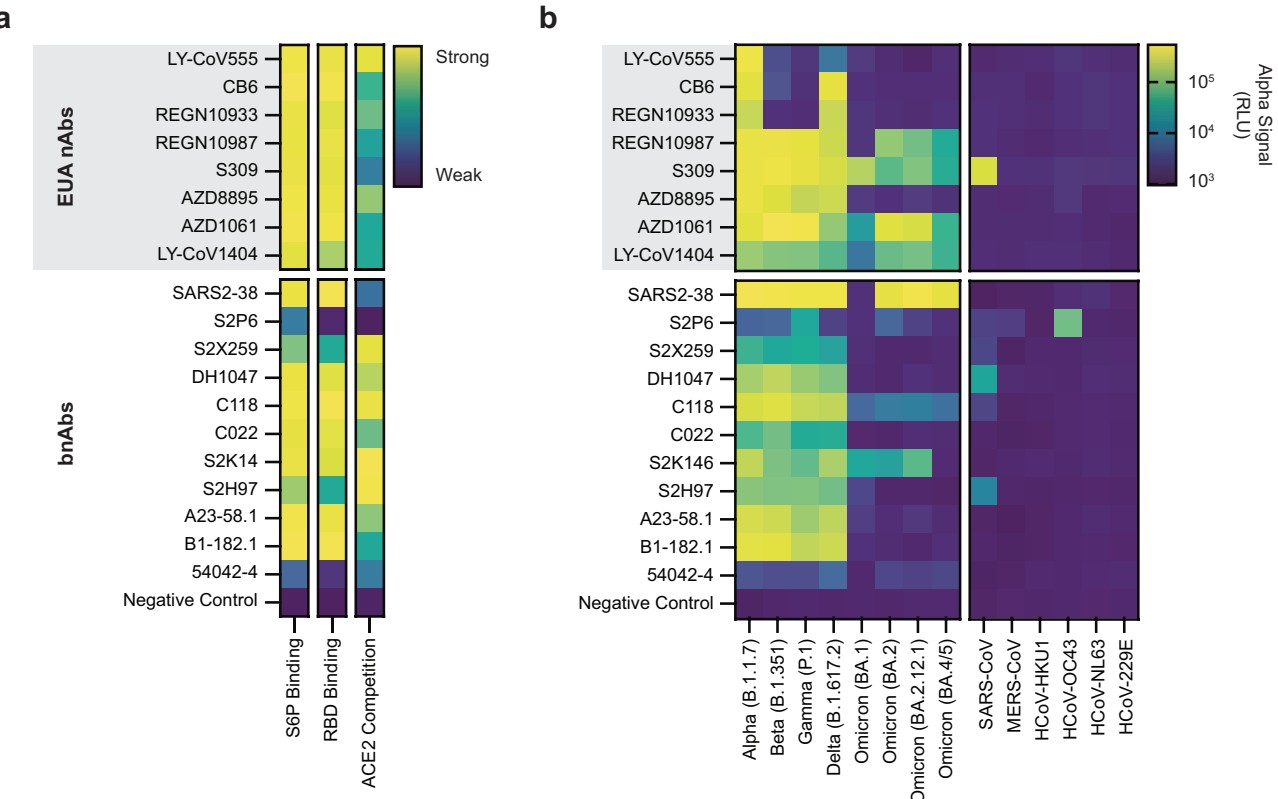

**Fig. 3 | Evaluating high interest COVID-19 neutralizing antibodies. a** Heatmaps of the binding of EUA and bnAb sdFabs measured using AlphaLISA to detect S6P binding (log$_{10}$ scaled), RBD binding (log$_{10}$ scaled), and ACE2 competition (linearly scaled). **b** Heatmap profiling binding of EUA and bnAb sdFabs against SARS-CoV-2 historical VOC and the coronaviruses SARS-CoV, MERS-CoV, HCoV-HKU1, HCoV-

OC43, HCoV-NL63, and HCoV-229E. Data are log$_{10}$ scaled. **a, b** AlphaLISA data are the mean of 3 replicates ($n = 3$) derived from independent CFPS reactions. Antigen concentrations for AlphaLISA experiments listed in Supplementary Table 2. Source data are provided in the Source data file.

antibodies, the normalized S6P binding AlphaLISA measurement correlates well with loss in neutralization potency (Two-sided Pearson correlation coefficients with 95% confidence interval and *p* value of *r* = 0.76 (0.63 to 0.84, *p* < 0.0001)). Also consistent with literature, S309 exhibited binding to the SARS-CoV spike protein, whereas no other EUA Ab is reported to bind SARS-CoV[9,54–59].

For the set of bnAbs, we observed cross-reactivity with the SARS-CoV spike protein for S2P6, S2X259, DH1047, C118, and S2H97 all of which are reported to bind to this antigen[10,61–64]. S2K146 did not exhibit binding to the SARS-CoV spike, despite being reported to neutralize SARS-CoV[65]. S2P6 additionally exhibited binding to MERS-CoV and HCoV-OC43 consistent with the literature on this antibody[61]. For S2P6, we observed heterogeneity in binding signal to different S6P variants, possibly a result of the target epitope of S2P6 being near the C-terminus of S6P and near to the C-terminal avi tag (a site-specific biotinylation), which may impede immobilization on the AlphaLISA bead. Neutralization data against all VOC for the tested bnAbs are not available, but the binding profiles for those characterized are generally consistent with literature. Against Omicron BA.1, S2K146 exhibits

strong binding whereas S2X259, S2H97, and SARS2-38 all exhibit reduced binding[69,70]. Similarly, S2K146 exhibits strong binding to all Omicron variants except BA.4/5, whereas S2X259 and S2H97 exhibit reduced binding to the other Omicron sub-lineages[71,72]. Taken together, our results indicate that the CFPS-derived antibody fragment binding patterns are consistent with those reported in literature.

## Discovery of SARS-CoV-2 antibodies from immunized mice

We next sought to discover antibodies against SARS-CoV-2 using our workflow. We immunized mice with ChAd-SARS-CoV-2-S expressing Wu-Hu-1 SARS-CoV-2 S2P[73] and isolated spike-positive activated B cells 10 days later using fluorescence-activated cell sorting. The pooled sorted B cells were sequenced to identify paired heavy and light chains, which were codon optimized and ordered commercially as synthetic DNA (Supplementary Data 2 and 3). We screened 119 identified antibody sequences measuring S6P binding, RBD binding, ACE2 competition, and sdFab assembly (Fig. 4). Like our initial screen with previously reported antibodies, a high consistency between independent experimental replicates was observed (Supplementary Fig. 11)

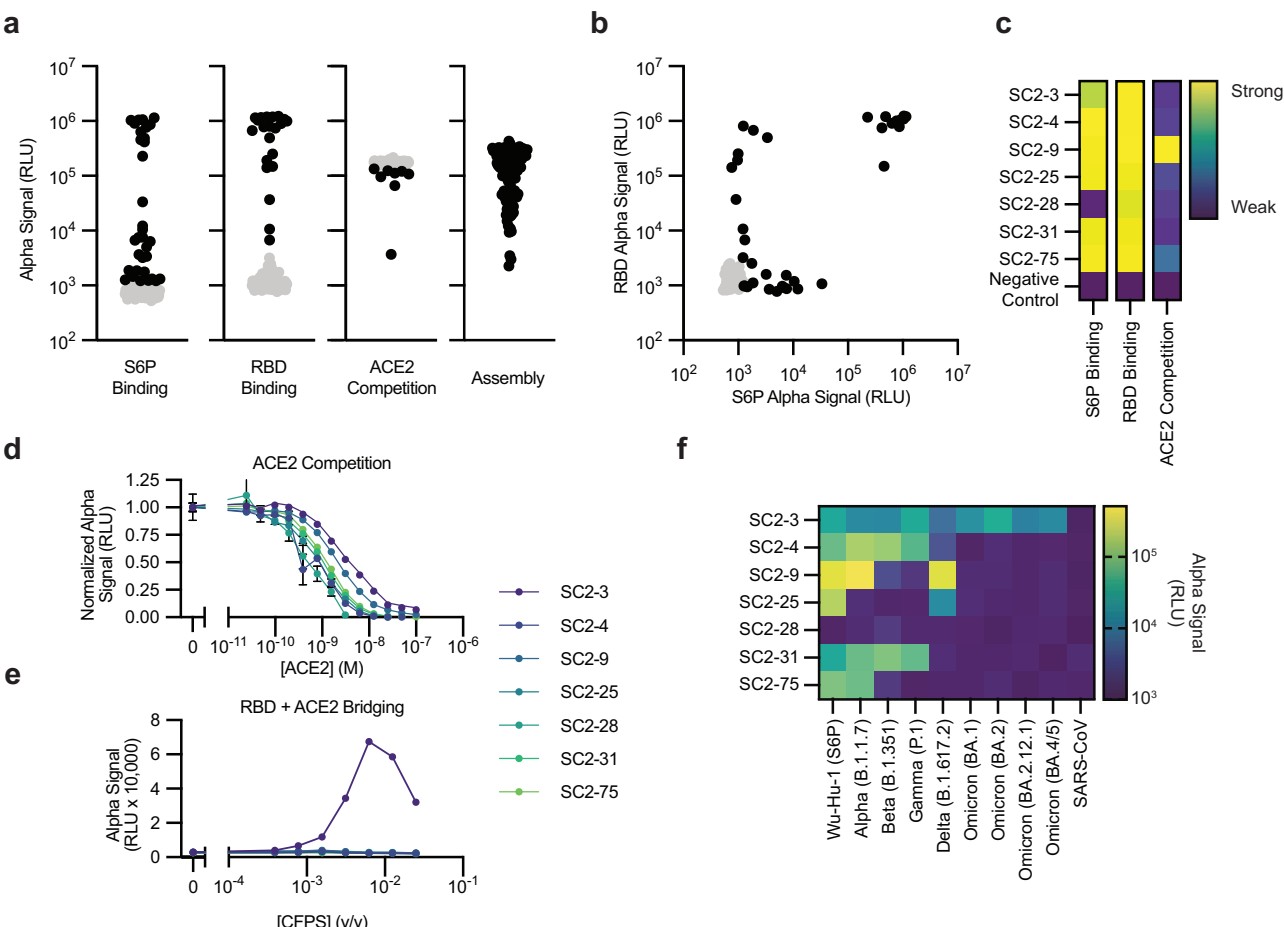

**Fig. 4 | Discovery of murine antibodies targeting SARS-CoV-2 using cell-free antibody screening. a** Summary of the S6P binding, RBD binding, ACE2 competition, and assembly AlphaLISA data for all 119 antibody fragments screened. **b** Comparison of RBD AlphaLISA data and S6P AlphaLISA data. **a, b** Black data points represent data that are statistically significant (two-sided t-test FDR corrected *p* < 0.05) from background and have an average value > 3 standard deviations away from background. Grey data were considered not significantly different from background. **c** Heatmaps of select antibody fragments detailing individual S6P binding, RBD binding, and ACE2 competition AlphaLISA data. S6P binding and RBD binding data are log₁₀ scaled and ACE2 competition is linearly scaled. **d** ACE2 competition measured by titrating the concentration of ACE2. sdFabs in crude

CFPS were diluted to a fixed concentration and combined with a fixed concentration of RBD and varying concentrations of ACE2. **e** ACE2 bridging by an sdFab and the RBD. A fixed concentration of ACE2 and the RBD was combined with varying concentrations of sdFab. **d–f** Error bars represent the standard error of the mean. An absence of error bars indicates error within the marker. **f** Heatmap profiling select antibodies for binding against SARS-CoV-2 historical VOC and SARS-CoV. Data are log₁₀ scaled. **a–f** AlphaLISA data are the mean of 3 replicates (*n* = 3) derived from independent CFPS reactions. Antigen concentrations for AlphaLISA experiments listed in Supplementary Table 2. Source data are provided in the Source data file.

with average coefficients of variation between 0.09 and 0.22 depending on the AlphaLISA measurement modality.

We observed assembly signal for all 119 antibody fragments and S6P or RBD binding for 37 of the screened antibody fragments (Fig. 4a), with some antibody fragments only exhibiting binding to either the RBD or S6P (Fig. 4b). Several of the antibody fragments competed with ACE2 in the initial screen (Fig. 4a, c and Supplementary Fig. 12), which we further validated via dose-response analysis (Fig. 4d). SC2-9 exhibited strong competition with ACE2 (required high concentrations of ACE2 for inhibition of binding to the RBD). Of note, the antibody SC2-3 required high concentrations of ACE2 to inhibit binding to the RBD and exhibited RBD bridging with ACE2 (Fig. 4e), like what was observed for S309 (Supplementary Fig. 9). Combined, these data suggest that SC2-3 either only weakly competes with ACE2 or that binding of SC2-3 to its target epitope may influence ACE2 binding. In the ACE2 bridging experiment, the reduction in AlphaLISA signal at higher concentrations likely is due to the "hook effect"[74] where binding sites on the AlphaLISA beads become saturated, and higher concentrations of antibody begin to inhibit signal. We also profiled the binding of these antibodies to historical SARS-CoV-2 VOC and SARS-CoV spike proteins (Fig. 4f). Most tested antibodies exhibited reduced binding to several different VOC spike proteins, except for SC2-3, which exhibited binding to all tested SARS-CoV-2 VOC spike proteins. These data indicate that our developed workflow can be used to discover antibodies that bind a target antigen.

## Discussion

Here we developed an integrated workflow for antibody fragment screening by combining methods for cell-free DNA assembly and amplification, cell-free protein synthesis, and binding characterization via AlphaLISA. This workflow has two key features. First, it is fast. The entire workflow can be completed in a matter of hours; indeed, the antibody screens reported in Figs. 2 and 4 were each a single experiment (from commercial synthetic DNA to data) completed in less than 24 h by a single researcher. Second, all workflow steps were developed with automation and throughput in mind. Each step in the process consists of straightforward liquid handling and temperature incubation steps. Although the workflow as reported here is not fully automated and requires manual intervention to move plates between liquid handling steps, end-to-end automation using a robotic arm and incubators will be straightforward due to the workflow's simplicity. Of note, the tools leveraged in this work are those that were developed for high-throughput small molecule drug discovery, a field where it is commonplace to evaluate $10^4$ to $10^5$ compounds in a screen[75]. Similar throughput should be attainable for high-throughput antibody screening using our reported methods with additional investment in automation.

Our workflow has several limitations and promising potential extensions. A limitation of the current workflow is that the sdFab antibody fragment format is not the final therapeutic format of the antibody, and thus sequences must be reformatted to full-length IgGs prior to scaled expression, be it cell- or cell-free-based, for further development. In addition, we utilized synthetic DNA coding for the antibodies of interest as opposed to PCR products from single B cells coding for the paired heavy and light chains. However, previous work suggests that this workflow is compatible with PCR products amplified from single B cells from an immunized animal[16,18,23]. Similarly, the workflow is agnostic to the method used for antibody selection, and thus is likely also compatible with other technologies beyond the isolation of single B cells like hybridomas and in vitro display techniques[76]. We also recently applied similar methods to those described here to engineer highly potent computationally designed minibinders that neutralize SARS-CoV-2[77], which suggests that this approach may be broadly applicable to different binding protein formats. Additional improvement of the CFPS system could allow for the use full-length antibodies during screening. Further development of AlphaLISA methods could enable high-resolution antigenic mapping of the immune response to antigens and antibody epitope binning via AlphaLISA-based competition or bridging experiments[78].

Overall, our rapid, cell-free workflow was successful in identifying and evaluating potent and drug-like neutralizing antibodies. While some weaker neutralizing antibodies were not completely characterized in our screen, we observed ACE2 competition for 14 of 17 antibodies in the overall data set whose neutralization $IC_{50}$ values are less than 0.01 µg/mL and whose mechanism is ACE2 competition (Figs. 2 and 3). Furthermore, our workflow identified all 8 historical EUA antibodies as well as most of the screened bnAbs. We also identified antibodies targeting the SARS-CoV-2 spike protein derived from an immunized mouse, including the antibody SC2-3, which binds to all tested historical SARS-CoV-2 VOC and warrants further investigation. Taken together, our data on CFPS-derived sdFabs are internally consistent and largely align with results in literature for the respective Fab or IgG molecules.

Looking forward, we anticipate that the increased speed and throughput afforded by our workflow will enable researchers to screen large numbers of antibodies easily and rapidly, facilitating down-selection to highly potent candidates that can be reformatted as IgGs, expressed at larger scales, and subjected to deeper development. In this way, our method is poised to aid in the discovery of robust medical countermeasures in future pandemics, and more broadly, in the development of binding proteins for therapeutic, diagnostic, and research applications.

## Methods

### Ethical statement

Animal studies were carried out in accordance with the recommendations in the Guide for the Care and Use of Laboratory Animals of the National Institutes of Health. The protocols were approved by the Institutional Animal Care and Use Committee at the Washington University School of Medicine (Assurance number A3381-01).

### Statistics and reproducibility

No statistical method was used to predetermine sample size. Previous experience with the measurement techniques and their dynamic ranges was used to determine sample sizes. Large-scale experiments (i.e., Figs. 2 and 4) were not replicated in a separate experiment in their entirety, but the top conditions were independently replicated in separate experiments with matching results. Other experiments were generally replicated in a separate experiment at least once, with the results matching the data reported in this work. The experiments were not randomized because no animal studies or clinical trials were performed to assess a medical intervention. The Investigators were not blinded to allocation during experiments and outcome assessment because no animal studies or clinical trials were performed. One replicate of a control sample in the data related to Fig. 4 was excluded from analysis due to a liquid handling error. The replicate is included in the Source Data and annotated as excluded.

Samples were considered different from background if they exhibited $p < 0.05$ on a two-sided t-test and had a value that was beyond the limit of detection (average background ± 3x standard deviation of the background measurement). The Benjamini and Hochberg False Discovery Rate procedure[50] was used to correct for multiple testing. Statistical analyses were performed in python. Two-sided t-tests (two-tailed, two sample, assumed equal variance) were performed using the scipy package and the FDR procedure was performed using the statsmodels package with a family-wise error rate of 5%. For antibody screening, the following samples were used as a measurement of background, and the combined data were used in the t-test. Assembly: No DNA and beads only controls. S6P binding: No

DNA and beads only controls. RBD binding: No DNA and beads only controls. ACE2 competition: No DNA and αHER2.

Pearson correlation coefficients were calculated using GraphPad Prism 9.

## Software

Single-cell BCR sequences were analyzed with Cell Ranger v3.1.0. AlphaLISA data were analyzed with Prism 9.5.1. Images were processed using ImageJ2 v2.9.0/1.53t. Plate reader data were processed using Python 3.8.8. The Python code to analyze plate reader data was not central to the research and was not deposited.

## Antibody sdFab sequence design

sdFabs were assembled based on a modified version of previously published protocols[16–18]. Example plasmid maps of the aHER2 heavy and light chain sdFabs can be found in the Source data. Antibody sequences were collected from literature and their light chains were classified as either kappa or lambda via the terminal residue of the J-segment in the VL domain. The VH and VL domains were subsequently fused to their corresponding human constant heavy (Uniprot P0DOX5) or human constant light (kappa CL Uniprot P01834 or lambda 1 CL Uniprot P0CG04) chains. At the N-terminus of the VH and VL domains, we chose to include a modified expression tag based on the first 5-residues of the *E. coli* chloramphenicol acetyltransferase gene followed by a Tobacco Etch Virus (TEV) protease cleavage site (protein sequence: MEKKIENLYFQS, DNA sequence: atggagaaaaaaatcgaaaacctgtacttccagagc)[79] as opposed to the previously published SKIK tag[80]. The heavy chain was fused to the LZA heterodimer subunit (AQLEKELQALEKENAQLEWELQALEKELAQK) and a strep II tag or super FLAG (sFLAG) tag[81]. The light chain was fused to the LZB heterodimer subunit (AQLKKKLQALKKKNAQLKWKLQALKKKLAQK). Antibodies in the screen in Fig. 2 were designed with the strep II tag on their heavy chain. Antibody sequences from the screens in Figs. 3 and 4 were designed with the sFLAG tag on their heavy chain. Antibodies in the screen in Fig. 2 and the EUA antibodies in Fig. 3 were designed with their native light chain class (kappa or lambda). Antibodies in the screen in Fig. 4 and the broadly neutralizing antibody sequences were designed with a kappa light chain, regardless of the native light chain class. Examples of the three types of antibody sequences are detailed below, with the important sequence features highlighted in square brackets [ ].

sdFab heavy chain constant strepII tagged:

[MEKKIENLYFQS][VH_Sequence][ASTKGPSVFPLAPSSKSTSGGTAALGCLVKDYFPEPVTVSWNSGALTSGVHTFPAVLQSSGLYSLSSVVTVPSSSLGTQTYICNVNHKPSNTKVDKKVEPKSC]GGGGS[AQLEKELQALEKENAQLEWELQALEKELAQK]GSSA[WSHPQFEK].

sdFab heavy chain constant super FLAG (sFLAG) tagged:

[MEKKIENLYFQS][VH_Sequence][ASTKGPSVFPLAPSSKSTSGGTAALGCLVKDYFPEPVTVSWNSGALTSGVHTFPAVLQSSGLYSLSSVVTVPSSSLGTQTYICNVNHKPSNTKVDKKVEPKSC]GGGGS[AQLEKELQALEKENAQLEWELQALEKELAQK]GSSA[DYKDEDLL].

sdFab light chain kappa:

[MEKKIENLYFQS][VL_Sequence][RTVAAPSVFIFPPSDEQLKSGTASVVCLLNNFYPREAKVQWKVDNALQSGNSQESVTEQDSKDSTYSLSSTLTLSKADYEKHKVYACEVTHQGLSSPVTKSFNRGEC]GGGGS[AQLKKKLQALKKKNAQLKWKLQALKKKLAQK].

sdFab light chain lambda 1:

[MEKKIENLYFQS][VL_Sequence][GQPKANPTVTLFPPSSEELQANKATLVCLISDFYPGAVTVAWKADGSPVKAGVETTKPSKQSNNKYAASSYLSLTPEQWKSHRSYSCQVTHEGSTVEKTVAPTECS]GGGGS[AQLKKKLQALKKKNAQLKWKLQALKKKLAQK].

## DNA assembly and linear expression template (LET) generation

Proteins to be manufactured via CFPS were codon optimized using the IDT codon optimization tool and ordered as double-stranded linear DNA containing the desired Gibson assembly overhangs from IDT or GenScript. sfGFP was ordered containing the two pJL1 Gibson assembly overhangs. Antibody VH DNA was ordered with the pJL1 5′ and the human IgG1 heavy chain constant 5′ Gibson overhangs. Antibody VL DNA was ordered with the pJL1 5′ and the human Ig light chain kappa or lambda 1 Gibson assembly overhangs. DNA was resuspended at a concentration of 50 ng/µL and used without amplification.

Additional linear DNA components for Gibson assembly (pJL1 backbone, sdFab heavy chain constant strepII tagged, sdFab light chain kappa constant, sdFab light chain lambda 1 constant) were ordered as gblocks from IDT. These components were amplified using PCR using Q5 Hot Start DNA polymerase (NEB, M0493L) following manufacturer instructions. Amplified DNA was purified using the DNA Clean and Concentrate Kit (Zymo Research, D4006) and diluted to a concentration of 50 ng/µL. Sequences of the utilized components are listed below, with Gibson assembly sequences being denoted by underlined lowercase text and primers for a given amplicon being listed below the DNA sequence.

Gibson assembly overhangs:

pJL1 5′ Gibson: tttgtttaactttaagaaggagatatacat.

pJL1 3′ Gibson: gtcgaccggctgctaacaaagcccgaaagg.

Human IgG1 heavy chain constant 5′ Gibson: gcgtcaacaaaggtccttcagttttcccattagcccct.

Human Ig light chain kappa 5′ Gibson: cgcacggtcgcggcgccgtctgtctttattttcctcct.

Human Ig light chain lambda 5′ Gibson: ggccaacccaaagcaaacccaactgtcactttgttcccg.

Linear pJL1 plasmid backbone (Addgene plasmid # 69496):

gtcgaccggctgctaacaaagcccgaaaggAAGCTGAGTTGGCTGCTGCCACCGCTGAGCAATAACTAGCATAACCCCTTGGGGCCTCTAAACGGGTCTTGAGGGGTTTTTTGCTGAAAGCCAATTCTGATTAGAAAAACTCATCGAGCATCAAATGAAACTGCAATTTATTCATATCAGGATTATCAATACCATATTTTTGAAAAAGCCGTTTCTGTAATGAAGGAGAAAACTCACCGAGGCAGTTCCATAGGATGGCAAGATCCTGGTATCGGTCTGCGATTCCGACTCGTCCAACATCAATACAACCTATTAATTTCCCCTCGTCAAAAATAAGGTTATCAAGTGAGAAATCACCATGAGTGACGACTGAATCCGGTGAGAATGGCAAAAGCTTATGCATTTCTTTCCAGACTTGTTCAACAGGCCAGCCATTACGCTCGTCATCAAAATCACTCGCATCAACCAAACCGTTATTCATTCGTGATTGCGCCTGAGCGAGACGAAATACGCGATCGCTGTTAAAAGGACAATTACAAACAGGAATCGAATGCAACCGGCGCAGGAACACTGCCAGCGCATCAACAATATTTTCACCTGAATCAGGATATTCTTCTAATACCTGGAATGCTGTTTTCCCGGGGATCGCAGTGGTGAGTAACCATGCATCATCAGGAGTACGGATAAAATGCTTGATGGTCGGAAGAGGCATAAATTCCGTCAGCCAGTTTAGTCTGACCATCTCATCTGTAACATCATTGGCAACGCTACCTTTGCCATGTTTCAGAAACAACTCTGGCGCATCGGGCTTCCCATACAATCGATAGATTGTCGCACCTGATTGCCCGACATTATCGCGAGCCCATTTATACCCATATAAATCAGCATCCATGTTGGAATTTAATCGCGGCTTCGAGCAAGACGTTTCCCGTTGAATATGGCTCATAACACCCCTTGTATTACTGTTTATGTAAGCAGACAGTTTTATTGTTCATGATGATATATTTTTATCTTGTGCAATGTAACATCAGAGATTTTGAGACACAACGTGAGATCAAAGGATCTTCTTGAGATCCTTTTTTTCTGCGCGTAATCTGCTGCTTGCAAACAAAAAAACCACCGCTACCAGCGGTGGTTTGTTTGCCGGATCAAGAGCTACCAACTCTTTTTCCGAAGGTAACTGGCTTCAGCAGAGCGCAGATACCAAATACTGTTCTTCTAGTGTAGCCGTAGTTAGGCCACCACTTCAAGAACTCTGTAGCACCGCCTACATACCTCGCTCTGCTAATCCTGTTACCAGTGGCTGCTGCCAGTGGCGATAAGTCGTGTCTTACCGGGTTGGACTCAAGACGATAGTTACCGGATAAGGCGCAGCGGTCGGGCTGAACGGGGGGTTCGTGCACACAGCCCAGCTTGGAGCGAACGACCTACACCGAACTGAGATACCTACAGCGTGAGCTATGAGAAAGCGCCACGCTTCCCGAAGGGAGAAAGGCGGACAGGTATCCGGTAAGCGGCAGGGTCGGAACAGGAGAGCGCACGAGGGAGCTTCCAGGGGGAAACGCCTGGTATCTTTATAGTCCTGTCGGGTTTCGCCACCTCTGACTTGAGCGTCGATTTTTGTGATGCTCGTCAGGGGGGCGGAGCCTATGGAAAAACGCCAGCAA

CGCGATCCCGCGAAATTAATACGACTCACTATAGGGAGACCACAACG
GTTTCCCTCTAGAAATAATtttgtttaactttaagaaggagatatacat.

pJL1_F: gtcgaccggctgcta.

pJL1_R: atgtatatctccttcttaaagttaaacaaaattatttcta.

Linear sdFab heavy chain constant strepII tagged:

gcgtcaa-
caaaaggtccttcagttttcccattagcccctTCTTCTAAGTCAACTAGTGGCGGTA
CTGCCGCTCTTGGGTGTTTGGTTAAAGATTACTTCCCAGAACCGGTT
ACGGTCTCGTGGAACTCTGGTGCACTGACATCGGGCGTACATACAT
TTCCCGCAGTTTTGCAGTCTTCGGGACTGTATTCTCTTTCATCGGTGG
TTACAGTCCCTAGCTCTTCCCTGGGTACACAGACCTACATTTGTAAT
GTTAATCATAAGCCGAGTAATACTAAGGTGGATAAAAAGGTGGAACC
GAAGTCTTGTGGTGGTGGCGGGTCAGCTCAACTGGAGAAGGAGTTAC
AGGCACTGGAAAAAGAGAATGCTCAACTTGAGTGGGAATTACAGGCG
TTAGAAAAAGAACTGGCCCAGAAGGGTTCTAGCGCATGGTCACATCC
CCAGTTCGAAAAATAAgtcgaccggctgctaacaaagcccgaaagg.

Linear sdFab heavy chain constant super FLAG tagged:

gcgtcaa-
caaaaggtccttcagttttcccattagcccctTCTTCTAAGTCAACTAGTGGCGGTA
CTGCCGCTCTTGGGTGTTTGGTTAAAGATTACTTCCCAGAACCGGTT
ACGGTCTCGTGGAACTCTGGTGCACTGACATCGGGCGTACATACATT
TCCCGCAGTTTTGCAGTCTTCGGGACTGTATTCTCTTTCATCGGTGG
TTACAGTCCCTAGCTCTTCCCTGGGTACACAGACCTACATTTGTAAT
GTTAATCATAAGCCGAGTAATACTAAGGTGGATAAAAAGGTGGAACC
GAAGTCTTGTGGTGGTGGCGGGTCAGCTCAACTGGAGAAGGAGTTA-
CAGGCACTGGAAAAAGAGAATGCTCAACTTGAGTGGGAATTACAGG
CGTTAGAAAAAGAACTGGCCCAGAAGGGTGGAGCCAGTCCAGCAGC
TCCTGCGCCTGGCGGGGACTACAAAGATGAAGACCTTCTTTAAgtcg
accggctgctaacaaagcccgaaagg.

IgGC_F: GCGTCAACAAAAGGTCCTTCAGTTTTC.

pJL1_3'Gib_R: CCTTTCGGGCTTTGTTAGCAGC.

Linear sdFab light chain kappa constant:

cgcacggtcgcggcgccgtctgtctttattttttcctcctTCTGATGAACAGCTTAA
ATCTGGGACAGCTTCTGTTGTATGTTTATTAAACAACTTTTACCCGCG
TGAGGCAAAAGTTCAATGGAAGGTAGACAACGCACTGCAAAGCGGAA
ATTCGCAGGAGTCAGTTACCGAACAGGATTCCAAGGATAGTACCTAC
TCCTTAAGTTCAACATTAACCCTGTCAAAGGCGGACTATGAAAAACA-
TAAGGTATATGCCTGCAAGTAACTCATCAGGGCTTATCATCCCCAG
TTACAAAATCTTTCAACCGTGGAGAATGCGGCGGCGGAGGTAGCGC
GCAGCTTAAGAAAAAATTGCAAGCCCTTAAAAAAAAAAAATGCCCAAC
TTAAATGGAAGCTGCAAGCCTTAAAAAAGAAATTGGCGCAGAAGTAA
gtcgaccggctgctaacaaagcccgaaagg.

kLC_F: TCGCGGCGCCGTCTG.

pJL1_3'Gib_R: CCTTTCGGGCTTTGTTAGCAGC.

Linear sdFab light chain lambda 1 constant:

ggccaacccaaagcaaacccaactgtcactttgttcccgCCCTCAAGCGAGGAA
CTTCAGGCTAATAAGGCCACGCTTGTTTGCCTGATCTCAGACTTTTA
TCCCGGTGCCGTAACAGTGGCTTGGAAGGCAGATGGTTCGCCGGTCA
AAGCGGGCGTGGAAACTACAAAGCCATCGAAACAGTCAAACAATAAA-
TATGCGGCATCAAGTTACTTGAGCCTTACCCCAGAACAGTGGAAGTC
ACACCGCTCGTACAGTTGTCAAGTTACACACGAGGGAAGTACAGTTG
AAAAGACCGTTGCCCCAACTGAATGTTCAGGCGGTGGTGGCTCAGC
GCAGTTAAAGAAAAAACTGCAGGCTTTGAAGAAAAGAATGCTCAAT
TAAAGTGGAAATTGCAGGCGTTGAAGAAGAAACTTGCGCAGAAGTAA
gtcgaccggctgctaacaaagcccgaaagg.

lLC_F: GGCCAACCCAAAGCAAACC.

pJL1_3'Gib_R: CCTTTCGGGCTTTGTTAGCAGC.

Linear sfGFP (same DNA sequence as Addgene Plasmid #102634). Note that the sequence of sfGFP is heavily modified and contains mutations from Bundy et al.[82].

tttgtttaactttaagaaggagatatacatATGAGCAAAGGTGAAGAACTGTTT
TACCGGCGTTGTGCCGATTCTGGTGGAACTGGATGGCGATGTGAACG
GTCACAAATTCAGCGTGCGTGGTGAAGGTGAAGGCGATGCCACGATT
GGCAAACTGACGCTGAAATTTATCTGCACCACCGGCAAACTGCCGGT
GCCGTGGCCGACGCTGGTGACCACCCTGACCTATGGCGTTCAGTGTT

TTAGTCGCTATCCGGATCACATGAAACGTCACGATTTCTTTAAATCTG
CAATGCCGGAAGGCTATGTGCAGGAACGTACGATTAGCTTTAAAGAT
GATGGCAAATATAAAACGCGCGCCGTTGTGAAATTTGAAGGCGATAC
CCTGGTGAACCGCATTGAACTGAAAGGCACGGATTTTAAAGAAGAT
GGCAATATCCTGGGCCATAAACTGGAATACAACTTTAATAGCCATAA
TGTTTATATTACGGCGGATAAACAGAAAAATGGCATCAAAGCGAATTT
TACCGTTCGCCATAACGTTGAAGATGGCAGTGTGCAGCTGGCAGATC
ATTATCAGCAGAATACCCCGATTGGTGATGGTCCGGTGCTGCTGCCG
GATAATCATTATCTGAGCACGCAGACCGTTCTGTCTAAAGATCCGAA
CGAAAAAGGCACGCGGGACCACATGGTTCTGCACGAATATGTGAATG
CGGCAGGTATTACGTGGAGCCATCCGCAGTTCGAAAAATAAgtcgaccg
gctgctaacaaagcccgaaagg.

Gibson assembly was used to assemble protein open reading frame DNA with the pJL1 backbone following the published protocol with the addition of 3.125 μg/mL of ET SSB (NEB, product no. M2401S)[83,84]. 20 ng of purified, linear pJL1 backbone, 20 ng of purified, linear sdFab VH or VL constant DNA, and 20 ng of the protein open reading frame insert were combined in 2 μL Gibson assembly reactions and incubated at 50 °C for 30 min. The unpurified assembly reactions were diluted in 40 μL of nuclease-free water (Fisher Scientific, AM9937) and 1 μL of the diluted reaction was used as the template for a PCR to generate linear expression templates (LETs) for CFPS. Linear expression templates were amplified via PCR using the pJL1_LET_F (ctgagatacctacagcgtgagc) and pJL1_LET_R (cgtcactcatggtgatttctcacttg) primers in a 50 μL PCR reaction using the Q5 Hot Start DNA polymerase (NEB, M0493L) following manufacturer instructions.

The DNA sequence of the *P. pyralis* luciferase containing a c-terminal strepII tag (fLuc, Uniprot Q27758) used as a negative control is below and was cloned into the pJL1 vector.

atggaagacgctaagaacattaagaagggacctgctccattctaccccctcgaagacgg
cactgcaggtgagcagcttcataaagcgatgaagcgttatgcgttagttcctggcacgatcgcc
ttcactgacgcgcacatcgaagtcaatatcacctacgctgaatactttgagatgagtgtgcgtct
ggcggaagccatgaagcgttatggccttaacacgaaccaccgcatcgttgtttgtagcgagaat
tccttacaattcttcatgcccgtccttggcgcgctgtttattggtgtggccgttgcaccagcaat
gacatctataatgagcgcgagttgttgaactccatgaacatttctcaaccaacagtggtgttcgt
ttcaaagaaaggcttacagaaaatcttaaacgttcaaaagaaactgccgattatccagaagatc
atcattatggatagtaagactgactaccagggcttccagtcaatgtatacattcgtgacgagtca
cctgcccccgggtttaacgagtacgactttgtcccagagagctttgatcgcgacaagaccat
cgccctcattatgaatagcagtggttcgacgggtagcccaaagggagtggccctgccccatcgt
accgcgtgcgtccgtttctcccatgcccgcgacccaatttttcggcaatcaaatcatccccgacac
ggcaatcttgtcggtcgtcccgtttcaccatggctttggaatgtttacgacactcggttacctcat
ctgcggtttccgcgtcgttctgatgtatcgcttcgaggaagagttgttcttacgttcgcttcagga
ctacaagattcaatccgcccttctggtccccactttgttcagtttctttgctaagagcaccttaatt
gataagtatgacctctccaacttacacgagattgcgagcggtggtgctcccctcagcaaagag
gttggagaggcggttgctaagcgttttcatctgccccggtatccgtcaaggttacggcctcaccg
aaaccacttctgccattcttatcactccggaaggtgacgataagcctggggcagtgggtaaag
ttgtaccctctctcgaggctaaggttgtggatttagatacggggaagaccttaggtgtgaacca
gcgcggtgaactgtgcgttcgcggtccgatgattatgtcggggtttatgttaatgaccccgaggc-
tacgaacgcgcttatcgataaggacggttggcttcattccggcgacatcgcttactgggatga
ggatgagcacttcttcatcgttgaccgtctgaagagtctcatcaagtataagggatgtcaagtc
gctccggcagagttagagagcatcttactccagcaccctaatatcttcgatgctggggttgccg
ggctcccaggcgacgatgccggcgagctgccggcggcggtagttgttttagagcatggcaag
accatgaccgaaaaggagattgtagactacgtcgcgagtcaagtaaccacagcgaagaagct
ccgcggtggagtggtctttgttgacgaggtgcctaaaggcctgacgggcaaacttgacgcgcg
taagatccgtgagatcctcatcaaagcgaagaaggtgggaagagtaagctggggagttcag
gttggtcccacccgcaatttgagaagtga.

### Cell extract preparation for cell-free protein synthesis

*E. coli* Origami™ B(DE3) (Novagen, 70837) extracts were prepared using a modified version of established protocols[85,86]. Briefly, a 150 mL Origami™ B(DE3) starter culture was inoculated in LB from a glycerol stock and cultured in a 250 mL baffled flask at 37 °C for 16 h. The 2xYTP was prepared without glucose in 75% of the final volume and sterilized using an autoclave. A 4x glucose solution was prepared and autoclaved separately, then added to the medium immediately before use. The

starter cultures were used to inoculate 1 L of 2xYTPG media (16 g/L tryptone, 10 g/L yeast extract, 5 g/L sodium chloride, 7 g/L potassium phosphate dibasic, 3 g/L potassium phosphate monobasic, 18 g/L glucose) in a 2.5 L Full-Baffle Tunair shake flask at an initial OD600 of 0.08. Cells were cultured at 37 °C at 220 RPM in a shaking incubator. Cultures were grown until OD600 0.4-0.6, at which point the expression of T7 RNA polymerase was induced by the addition of IPTG to a final concentration of 0.5 mM. Cells were harvested at an OD600 of 2.5 via centrifugation at $12,000 \times g$ for 1 min at 4 °C. Cell pellets were washed three times with 25 mL S30 buffer per 50 mL culture (10 mM Tris Acetate pH 8.2, 14 mM Magnesium Acetate, and 60 mM Potassium Acetate). Pellets were resuspended in 1 mL S30 buffer per gram of cell mass. Cell suspensions were lysed using a single pass on an Avestin EmulsiFlex-B15 Homogenizer at a lysis pressure of 24,000 PSI. Cell debris was separated via centrifugation at $18,000 \times g$ for 20 min, and the clarified lysate was collected, flash-frozen in liquid nitrogen, and stored at −80 °C.

## Cell-free protein synthesis reactions

CFPS reactions were composed of the following reagents: 8 mM magnesium glutamate, 10 mM ammonium glutamate, 130 mM potassium glutamate, 1.2 mM ATP, 0.5 mM of each CTP, GTP, and UTP. 0.03 mg/mL folinic acid, 0.17 mg/mL *E. coli* MRE600 tRNA (Roche 10109541001), 100 mM NAD, 50 mM CoA, 4 mM oxalic acid, 1 mM putrescine, 1 mM spermidine, 57 mM HEPES pH 7.2, 2 mM of each amino acid, 33.3 mM PEP, 20% v/v *E. coli* extract, varying concentrations of DNA template, and the remainder water. The preparation of these reagents has been described in detail elsewhere[87]. For DNA templates, plasmids were used at a concentration of 8 nM, and unpurified linear PCR products were used at 6.66% (v/v). For the expression of antibodies, each template was added to a final concentration of 6.66% (v/v). For antibody and sdFab expression 4 mM oxidized glutathione, 1 mM reduced glutathione, 14 μM of purified DsbC, and 50 μM FkpA were also supplemented to the reactions. In addition, for oxidizing CFPS reactions, cell-extracts were treated with 500 μM iodoacetamide (IAM) at room temperature for 30 min before use in CFPS[88]. All reaction components were assembled on ice and were either run as 12 μL reactions in 1.5 mL microtubes or 2 μL reactions in 384-well plates (BioRad, HSP3801). For 2 μL reactions, components were transferred to the plate using an Echo 525 acoustic liquid handler. A mix containing all the CFPS components except for the DNA was dispensed from 384PP Plus plates (Labcyte, PPL-0200) using the BP setting. The DNA (unpurified PCR products) was dispensed from a 384LDV Plus plate (Labcyte, LPL-0200) using the GP setting. Reactions were allowed to proceed at 30 °C for 20 h.

## Quantification of cell-free protein synthesis reactions

To quantify sfGFP fluorescence, a standard curve was prepared using previously reported methods[86]. Radioactive leucine was added to CFPS at a final concentration of 10 μM of L-[14C(U)]-leucine (Perkin Elmer NEC279E250UC, 11.1GBq/mMole), followed by precipitation of the expressed proteins and scintillation counting[89]. To quantify sfGFP fluorescence, 2 μL of a CFPS reaction was diluted in 48 μL of water in a Black Costar 96 Well Half Area Plate. Fluorescence was measured using a BioTek Synergy™ H1 plate reader with excitation and emission wavelengths of 485 and 528, respectively. Scintillation counts and fluorescence were fit to determine a standard curve for use with non-radioactive samples.

To visualize antibody assembly, proteins were labeled during CFPS with FluoroTect™ (Promega, L5001). FluoroTect™ was included in the CFPS reaction at 3.33%v/v. After protein synthesis, RNAseA (Omega Bio-Tek, AC118) was added to 0.1 mg/mL and the sample was incubated at 37 °C for 10 min. 3 μL of the CFPS and RNAseA mixture were combined with 4x loading buffer (LiCor, 928-40004) and the samples were subsequently denatured at 70 °C for 3 min, then

separated via SDS-PAGE and imaged using a LI-COR Odyssey Fc imager on the 600 channel. Densitometry was performed using the ImageJ software.

## DsbC and FkpA expression and purification

Protein expression, purification, and his tag removal were performed similarly to previously reported[77]. DsbC (Uniprot P0AEG6, residues 21–236) and FkpA (Uniprot P45523, residues 26–270) were ordered as gBlocks from IDT containing a c-terminal, TEV cleavable his tag (GSENLYFQSGSHHHHHHHHHH) and cloned into pET28a. Plasmid maps of both DsbC and FkpA are available in the Source Data. Plasmids were transformed into BL21 Star™ DE3, plated on LB agar, and cultured overnight at 37 °C. 1 L of Overnight Express TB (Fisher Scientific, 71491-4) was inoculated by scraping all colonies on a transformation plate and cultured at 37 °C in 2.5 L tunair flasks (IBI Scientific, SS-8003) at 220 rpm overnight. Cells were harvested, resuspended at a ratio of 1 g cell mass to 4 mL resuspension buffer (50 mM HEPES pH 7.5, 500 mM NaCl, 1X HALT protease inhibitor without EDTA (Fisher Scientific, 78429), 1 mg/mL lysozyme, 62.5 U/mL cell suspension of benzonase (Sigma-Aldrich, E1014-25KU)) and lysed using an Avestin B15 homogenizer at 24,000 PSI. The lysate was spun down $14,000 \times g$ for 10 min and the clarified supernatant was incubated with Ni-NTA Agarose (Qiagen, 30230) for 60 min on an end-over-end shaker. The resin was spun down $2500l \times g$ for 2 min, the supernatant removed, resuspended in wash buffer (50 mM HEPES pH 7.5, 500 mM NaCl, 50 mM Imidazole), loaded on a gravity flow column, and subsequently washed with 20X resin volumes of wash buffer. Protein was eluted using elution buffer (50 mM HEPES pH 7.5, 500 mM NaCl, 500 mM Imidazole) and exchanged into 50 mM HEPES pH 7.4, 150 mM NaCl using PD-10 desalting columns (Cytvia, 17-0851-01) according to manufacturer instructions.

His tags were removed via cleavage by ProTEV Plus (Promega, V6102). Before cleavage, 10% v/v glycerol was added to the protein. ProTEV Plus was added to a concentration of 0.5 U/μg purified protein and DTT was added to a concentration of 1 mM. Cleavage reactions were carried out at 30 °C for 4 h. Free His tag and ProTEV Plus were removed by incubating with Ni-NTA Agarose for 1 h at 4 °C and collecting the supernatant. Proteins were subsequently concentrated to >1 mg/mL (Millipore, UFC800396). His tag removal was validated via SDS-PAGE and the AlphaScreen Histidine (Nickel Chelate) Detection Kit (Perkin Elmer, 6760619C).

## AlphaLISA reactions

AlphaLISA reactions were carried out in 50 mM HEPES pH 7.4, 150 mM NaCl, 1 mg/mL BSA, and 0.00015 v/v TritonX-100 (hereafter referred to as Alpha buffer). All components were dispensed using an Echo 525 liquid handler from a 384-Well Polypropylene 2.0 Plus microplate (Labcyte, PPL-0200) using the 384PP_Plus_GPSA fluid type. All components of the AlphaLISA reactions were prepared as 4x stocks and added as 0.5 μL to the final 2 μL reaction to achieve the desired concentration. All AlphaLISA reactions were performed with CFPS reactions diluted to a final concentration of 0.025 v/v. AlphaLISA beads were combined to prepare a 4X stock in Alpha buffer immediately before use and added to the proteins to yield a concentration of 0.08 mg/mL donor beads and 0.02 mg/mL acceptor beads in the final reaction. All reactions were incubated with AlphaLISA beads for at least 1 h before measurement. AlphaLISA measurements were taken on a Tecan Infinite M1000 Pro plate reader using the AlphaLISA filter with an excitation time of 100 ms, an integration time of 300 ms, and a settling time of 20 ms. Before measurement, plates were allowed to equilibrate inside the instrument for 10 min. For measurements involving sdFabs, protein A AlphaLISA beads were avoided due to the ability of protein A to bind human subgroup VH3 Fabs[90].

The impact of CFPS reagents on AlphaLISA was determined by serially diluting the specified reagents in Alpha buffer and combining

them with the specified AlphaLISA conditions. The TrueHits kit (Perkin Elmer, AL900) was used to assess the impact of the CFPS reagents on the Alpha detection chemistry. CFPS reagents were mixed with the donor and acceptor beads and incubated for 2 h before measurement. His tagged RBD (Sino Biological, 40592-V08H) and human FC tagged human ACE2 (GenScript, Z03484) were used to evaluate the impact of CFPS reagents on capture chemistries. RBD and ACE2 were diluted in Alpha buffer, mixed at a final reaction concentration of 10 nM each, combined with the CFPS reagents, and allowed to incubate for 1 h. Donor and acceptor beads were subsequently added and allowed to incubate for a further hour before measurement. Protein A Alpha donor beads (Perkin Elmer, AS102), Ni-Chelate AlphaLISA acceptor beads (Perkin Elmer, AL108), and anti-6xhis AlphaLISA acceptor beads (Perkin Elmer, AL178) were utilized for detection.

The commercial neutralizing antibody ACE2 competition experiment was performed with the following antibodies: nAb1 (Acro Biosystems, SAD-S35), nAb2 (Sino Biological, 40592-MM57), nAb3 (Sino Biological, 40591-MM43), nAb4 (Sino Biological, 40592-R001). ELISA $IC_{50}$ values were recorded from the product page at the time of purchase and converted to µg/mL assuming a MW of 150,000 Da if reported in M. Antibodies were serially diluted in Alpha buffer and mixed with SARS-CoV-2 RBD (Sino Biological, 40592-V02H) at a concentration of 10 nM in the final reaction and incubated for 1 h. Mouse FC tagged human ACE2 (Sino Biological, 10108-H05H) was subsequently added and incubated for 1 h, followed by simultaneous addition of the acceptor and donor beads. AlphaLISA detection was performed using Anti-Mouse IgG Alpha Donor beads (PerkinElmer, AS104) and Strep-Tactin AlphaLISA Acceptor beads (PerkinElmer, AL136). $IC_{50}$ values were calculated using Prism 9 by fitting the normalized data to [Inhibitor] vs. response–Variable slope (four parameters) fit with the max constrained to a value of 1.

For all antibody screening experiments, the different reagents and AlphaLISA conditions used are described in Supplementary Table 2. The different AlphaLISA measurements were carried out as described below.

Assembly AlphaLISA reactions consisted of sdFab expressing CFPS and either Rabbit Anti-Human kappa light chain antibody (Abcam, ab125919) or Rabbit Anti-Human lambda light chain (Abcam, ab124719). CFPS reaction containing the expressed sdFab of interest was mixed with the appropriate anti-light chain antibody and allowed to equilibrate for two hours before the simultaneous addition of the acceptor and donor beads.

SARS-CoV-2 S6P binding AlphaLISA reactions consisted of sdFab expressing CFPS and SARS-CoV-2 S6P CFPS reaction containing the expressed sdFab of interest was mixed with the S6P and allowed to equilibrate for two hours before the simultaneous addition of the acceptor and donor beads.

SARS-CoV-2 RBD binding AlphaLISA reactions consisted of sdFab expressing CFPS and SARS-CoV-2 RBD. CFPS reaction containing the expressed sdFab of interest was mixed with the RBD and allowed to equilibrate for two hours before the simultaneous addition of the acceptor and donor beads.

ACE2 and RBD competition AlphaLISA reactions consisted of sdFab expressing CFPS, human ACE2, and SARS-CoV-2 S6P. CFPS reaction containing the expressed sdFab of interest was first mixed with S6P and allowed to incubate for 1 h. Subsequently, ACE2 was added and allowed to equilibrate for a further 1 h before the simultaneous addition of the acceptor and donor beads.

For SARS-CoV-2 variant and other non-SARS-CoV-2 coronavirus binding experiments, AlphaLISA measurements were carried out in the same manner as described for SARS-CoV-2 S6P. The following Hisx6-tagged proteins were used. SARS-CoV-2 S6P (Acro Biosystems, SPN-C52H9), SARS-CoV-2 S6P Alpha/ B.1.1.7 (Gift from Lauren Carter at the Institute for Protein Design at the University of Washington, expressed and purified as described elsewhere[77]), SARS-CoV-2 S6P Beta/B.1.351

(Gift from Lauren Carter at the Institute for Protein Design at the University of Washington, expressed and purified as described elsewhere[77]), SARS-CoV-2 S6P Gamma/P.1 (Gift from Lauren Carter at the Institute for Protein Design at the University of Washington, expressed and purified as described elsewhere[77]), SARS-CoV-2 S6P Delta/B.1.617.2 (AcroBiosystems, SPN-C52He), SARS-CoV-2 S6P Omicron/BA.1 (AcroBiosystems, SPN-C52Hz), SARS-CoV-2 S6P Omicron/BA.2 (AcroBiosystems, SPN-C5223), SARS-CoV-2 S6P Omicron/BA.2.12.1 (AcroBiosystems, SPN-C522d), SARS-CoV-2 S6P Omicron/BA.4/5 (AcroBiosystems, SPN-C522e), SARS-CoV S2P (AcroBiosystems, SPN-S52H6), MERS-CoV S2P (AcroBiosystems, SPN-M52H4), HCoV-HKU1 S (AcroBiosystems, SPN-H52H5), HCoV-OC43 S (AcroBiosystems, SPN-H52Hz), HCoV-NL63 S (AcroBiosystems, SPN-H52H4), and HCoV-229E S (AcroBiosystems, SPN-H52H3).

In the dose-dependent ACE2 competition titration experiments CFPS reactions were incubated with SARS-CoV-2 RBD for 1 h followed by the addition of the specified concentration (two-fold serially diluted from 100 nM) of human ACE2. All three components were incubated for an additional hour prior to simultaneous addition of AlphaLISA beads. Reactions were incubated for 2 h prior to measurement.

For RBD and ACE2 bridging experiments SARS-CoV-2 RBD, human ACE2, and the specified dilution of CFPS (two-fold serially diluted from 0.025 v/v) were incubated for 1 h prior to the simultaneous addition of the AlphaLISA beads. Reactions were incubated for 2 h prior to measurement.

### Mouse Immunization, cell staining, and sorting

Female C57BL/6 (Strain: 000664) were purchased from The Jackson Laboratory. Six-week-old animals were immunized with $10^{10}$ viral particles (vp) of ChAd-SARS-CoV-2-S[73] in 50 µl of PBS via intramuscular injection in the hind leg. Draining inguinal lymph nodes were collected 10 days later and processed into a single-cell suspension. Cells were stained with biotinylated recombinant SARS-CoV-2 spike (S2P) for 30 min at 4 °C then washed twice with FACS buffer followed by staining with anti-CD19 BV421 (BioLegend # 115537), anti-CD4 FITC (BioLegend # 100405), anti-IgD-PE-Cy7 (BioLegend # 405719), Streptavidin APC (BioLegend # 405207), aqua cell viability dye (Invitrogen L34957), and anti-mouse CD16/CD32 Fc block (BioLegend # 156607). Spike-positive activated B cells (live singlet CD4- CD19+ IgDlo Streptavidin+) were bulk sorted on BD FACSAriaII sorter.

### Single-cell RNA-seq library preparation and sequencing

The following 10x Genomics kits were used for libraries preparation: Chromium Single Cell 5' Library and Gel Bead Kit v2 (PN-1000006), Chromium Single Cell A Chip Kit (PN-1000152), Chromium Single Cell V(D)J Enrichment Kit, Mouse B cell (96rxns) (PN-1000072), and Single Index Kit T (PN-1000213). The GEM generation and barcoding was followed by cDNA preparation then GEM RT reaction and bead cleanup steps. Purified cDNA was amplified for 10–14 cycles then cleaned up using SPRIselect beads. cDNA concentration was determined by running samples on a Bioanalyzer. BCR target enrichments were done on the full-length cDNA followed by BCR libraries preparation as recommended by 10x Genomics Chromium Single Cell V(D)J Reagent Kits (v1 Chemistry) user guide. The cDNA Libraries were sequenced on Novaseq S4 (Illumina), targeting a median sequencing depth of 5000 read pairs per cell.

### Processing of single-cell BCR sequences

Demultiplexed pair-end FASTQ reads from 10x Genomics single-cell V(D)J profiling were preprocessed using the "cellranger vdj" command from *Cell Ranger* v3.1.0 for alignment against the GRCm38 mouse reference v3.1.0 (*refdata-cellranger-vdj-GRCm38-alts-ensembl-3.1.0*), generating 3760 assembled high-confidence BCR sequences for 4420 cells. Sequences for screening were selected randomly from the top clonal groups with >10 members in the clonal group. Cellranger vdj

output was then parsed using Change-O v0.4.6 within the immcantation suite. Additional quality control included examining sequences to be productively rearranged and have valid V and J gene annotations. Furthermore, only cells with exactly one heavy chain sequence paired with at least one light chain sequence were kept.

### Reporting summary

Further information on research design is available in the Nature Portfolio Reporting Summary linked to this article.

## Data availability

The data generated in this study are available in the Source data. Protein and DNA sequences for all antibodies expressed in this work are available in Supplementary Data 2. Original cDNA sequences for antibodies derived from immunized mice originating from this work are available in Supplementary Data 3 and have also been deposited to GenBank (Accession Numbers OQ570981–OQ571099 for VH sequences and OQ571100–OQ571218 for VL sequences). The raw sequencing data have been deposited to the Sequence Read Archive under the accession number PRJNA974195. Source data are provided with this paper.

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

## Acknowledgements

We thank Kosuke Seki at Northwestern University for providing the fLuc plasmid and Lauren Carter and Rashmi Ravichandran at the University of Washington Institute for Protein Design for providing the purified Alpha, Beta, and Gamma SARS-CoV-2 S6P proteins. M.C.J. acknowledges support from the David and Lucile Packard Foundation, the Camille Dreyfus Teacher-Scholar Program, and the Defense Threat Reduction Agency Grants HDTRA1-20-1-0004 and HDTRA1-21-1-0038. M.S.D. acknowledges the support of NIH grant R01 AI157155. A.C.H. was supported by a Department of Defense National Defense Science and Engineering Graduate (NDSEG) Fellowship (NDSEG-36373). L.G. was supported by a National Science Foundation Graduate Research Fellowship (DGE-2234667).

## Author contributions

A.C.H., B.V., and M.C.J. conceived of the study. A.C.H., B.V., A.H., L.G., A.K.G., and M.D. completed the research. W.K. and D.J.Y. assisted with cell-free expression methods. A.O.H. performed the B cell sorting and antibody characterization experiment in mice. A.S.K., M.S.D., and M.C.J. supervised the research. A.C.H. wrote the manuscript. All authors commented on and edited the manuscript.

## Competing interests

A.C.H. has consulted for SwiftScale Biologics, National Resilience Inc., and L.E.K Consulting. W.K.K. and D.J.Y. are current employees of National Resilience Inc. M.C.J. is a cofounder of SwiftScale Biologics, Stemloop, Inc., Design Pharmaceuticals, and Pearl Bio. M.C.J.'s interests are reviewed and managed by Northwestern University in accordance with their conflict-of-interest policies. M.S.D. is a consultant for Inbios, Vir Biotechnology, Senda Biosciences, Moderna, and Immunome. The Diamond Laboratory has received unrelated funding support in sponsored research agreements from Vir Biotechnology, Emergent BioSolutions, and Moderna. All other authors declare no competing interests.
