## [Peer Review File · Nature Communications]

REVIEWER COMMENTS

Reviewer #1 (Remarks to the Author):

This paper describes a new work-flow for therapeutic antibody discovery using cell free systems and alpha-ELISA screening with validation using a panel of published neutralising anti-COVID19 antibodies. The authors state that the advantage in their methods is the speed of multi-parallel antibody expression in plate format and screening. Although there is novelty in the methods for antibody VH and VL gene assembly / construct generation and expression, the screening techniques are not particularly novel. The work and methods described has some interest but I do not think will be widely adopted in the therapeutic antibody discovery community because the sdFab or “zip-body” format is not the final therapeutic format and so any hits from this pipeline would need to be re-formatted and re-tested. The authors do not mention the additional work involved to do this. The antibody community are moving more to high throughput final format screening which also has the advantage that developability can be screened at an earlier stage. Therefore this reviewer does not believe the methods will be widely adopted for future therapeutic antibody discovery by the main practitioners in the field, so it can be questioned if the work can be considered for a high impact journal such as Nature Communications. Submission to a specialised journal may be more appropriate. Detailed comments are listed below:

1. Line 105 – reference the source of the pJL1 vector.
2. Line 117 – ensure that correct reference and detailed production methods of the in vitro translation mixture is detailed in the methods section.
3. Supplementary Figure 1 line 38 – describe GFP quantitation.
4. Supplementary Figure 2a – minus DTT – quantitate the % of fully intact disulphide linked IgG 4 chain species. There appears to be some heterogeneity. Line 120 in main text (assembly of full IgG) is not supported by the SDS-PAGE result
5. Supplementary Figure 2b/c – was this performed plus or minus DTT?. Different antibodies showed different levels of oxidised product – why?
6. Line 127 – “more consistent expression of zip-bodies” is not supported by Supplementary Figure 2c where there is some heterogeneity in intact product.
7. Supplementary Figure 3e – why does the Regeneron antibody show a low signal although the expression level looked OK in Supp. Fig 2c?
8. Supplementary Figure 4 requires a greater explanation as to what is meant by the y-axis “number of constructs” and the aim of this experiment.

Reviewer #2 (Remarks to the Author):

In this article, Hunt et al. report on the use of automated CFPS to enhance the throughput of antibody discovery. The ability of cell-free synthesis that can produce recombinant proteins without involving cell cultivation steps is truly useful for streamlining the discovery and engineering of novel proteins. In particular, the availability of high-throughput PPI assay methods, including the Alpha Screen used in this study, makes it more feasible to build a completely “cell-free” process of antibody discovery by implementing automated CFPS. Authors successfully automated the steps of cell-free expression and analysis of anti-SARS-CoV-2 antibodies. Presented results support the possibility of using the proposed approach to cope with emerging pandemic, which will draw attention of the broad scientific community.

However, I would like to see several points clarified before its publication in Nature Communications.

1. Difficulties in proper folding and assembly of Fab have been a major issue in cell-free synthesis of antibodies. Authors utilized AlphaLISA to monitor antibody expression and assembly in CFPS, which is bit confusing. In their experiment, AlphaLISA for Fab assembly was designed to produce signal when the Fab chains on the donor bead and the acceptor bead are assembled to bring the two different beads in close proximity. How can it be assured that the assembly took place only between the Fab chains on different beads? Assembly of Fab chains on the same bead would not produce the AlphaLISA signal, and thus would lead to wrong evaluation of the targeted antibodies.

2. The size of antibody library is not large enough to demonstrate the impact of the proposed approach. In fact, the set of 120 antibodies can be readily expressed and analyzed by conventional manual CFPS reactions. It would be desirable to expand the size of clonal library to justify the implementation of automated procedures.

3. Are there specific reasons that authors used different liquid handling equipment for Gibson assembly and CFPS? Especially when considering the number of antibodies tested, I am not sure about the advantage of combining conventional liquid handling robot and the Echo system.

4. While authors performed their work with a collection of cloned genes, it will make their claims more convincing if they can demonstrate the proposed workflow can be adopted for the discovery of antibodies starting from B cells, which will require the integration of single cell RT-PCR step.

Reviewer #3 (Remarks to the Author):

In this manuscript, the authors describe an integrated pipeline for antibody expression and characterization, including four key developments. The high-throughput and cell-free antibody screening workflow was validated using 120 published antibodies.

Major comments:

1. This study used “previously published antibodies” to show the platform is functional. However, the authors should apply the platform in a discovery screen and demonstrate that it can be used to rapidly identify “novel” neutralizing antibodies against SARS-CoV-2 or other pathogens from B cells of convalescent patients within several hours.

General comments:

1. As the authors mention, this workflow has advantages of higher speed and throughput when performing discovery screens for antibodies from single B cells. Can this platform also be adapted for use with hybridoma technologies or phage display libraries?
2. The authors show a correlation between assembly, protein-protein interactions and competition. For cases such as REGN10987, which had low Alpha signal of assembly and RBD binding, would the antibody be selected in an actual high throughput screen?
3. An AlphaLISA method is used to characterize protein-protein interactions and neutralizing antibodies. The authors stated that decreased AlphaLISA signals indicate interference with protein-protein interactions; however, in Supplementary Figure 3c, the higher concentration of “cell lysate” sample had a higher signal. Do increases in signal have some meaning such as improvement of protein-protein interaction?
4. Does protein aggregation affect the signal of the AlphaLISA (leading to false positives)? Did aggregation occur in this study? How do the authors rule out this possibility?
5. In Figure 1b-d, the authors mentioned that they tested five different commercial antibodies (Line 150 of page 8), but in Figure 1c and d, there are only four antibodies shown.
6. Statistical tests are needed for Figure 2c-f.
7. “The Salt Solution and Amino Acids Mix inhibit immobilization of his-tagged proteins on the NiChelate bead.” in legend of Supplementary Figure 3c is redundant.
8. The authors need to describe how they determined the concentration and percentage of assembled sFab in the AlphaLISA system.

REVIEWER COMMENTS

Reviewer #1 (Remarks to the Author):

This paper describes a new work-flow for therapeutic antibody discovery using cell free systems and alpha-ELISA screening with validation using a panel of published neutralising anti-COVID19 antibodies. The authors state that the advantage in their methods is the speed of multi-parallel antibody expression in plate format and screening. Although there is novelty in the methods for antibody VH and VL gene assembly / construct generation and expression, the screening techniques are not particularly novel. The work and methods described has some interest but I do not think will be widely adopted in the therapeutic antibody discovery community because the sdFab or “zip-body” format is not the final therapeutic format and so any hits from this pipeline would need to be re-formatted and re-tested. The authors do not mention the additional work involved to do this. The antibody community are moving more to high-throughput final format screening which also has the advantage that developability can be screened at an earlier stage. Therefore this reviewer does not believe the methods will be widely adopted for future therapeutic antibody discovery by the main practitioners in the field, so it can be questioned if the work can be considered for a high impact journal such as Nature Communications. Submission to a specialised journal may be more appropriate. Detailed comments are listed below:

We appreciate the reviewer highlighting the broad interest of our method and providing us an opportunity to clarify the impact of the work.

While we agree that high-throughput protein-protein interaction analysis tools like AlphaLISA are well established, the novelty of our work is that when AlphaLISA and CFPS are combined, it enables high-throughput antibody functional screening in an individually addressable format on the scale of small molecule drug discovery. We believe this approach will serve as a tool to increase the quantity of quality hits (by enabling the screening of many more antibody sequences) and decrease the effort required to identify, individually study/manipulate, and obtain quantitative data on each of those hits. After we have identified hits, they can be reformatted as a full-length IgG (or whichever other format is desired) and evaluated using more labor-intensive methods. Building upon these ideas Reviewer 2 writes that the approach is “truly useful for streamlining the discovery and engineering of novel proteins. In particular, the availability of high-throughput PPI assay methods, including the Alpha Screen used in this study, makes it more feasible to build a completely “cell-free” process of antibody discovery by implementing automated CFPS.”

In the revised manuscript, we now directly address the need to reformat antibodies as a limitation in the discussion section:

“A limitation of the current workflow is that the sdFab antibody fragment format is not the final therapeutic format of the antibody, and thus sequences must be reformatted to full-

length IgGs prior to scaled expression, be it cell- or cell-free-based, for further development.”

“Looking forward, we anticipate that the increased speed and throughput afforded by our workflow will enable researchers to screen large numbers of antibodies easily and rapidly, facilitating down-selection to highly potent candidates that can be reformatted as IgGs, expressed at larger scales, and subjected to deeper development.”

Furthermore, we refer to our screened antibody molecules as antibody fragments of sdFabs wherever appropriate.

1. Line 105 – reference the source of the pJL1 vector.

Thank you for pointing out this oversight. We have now referenced the addgene part number (Addgene Plasmid #69496) in the relevant methods section.

2. Line 117 – ensure that correct reference and detailed production methods of the in vitro translation mixture is detailed in the methods section.

We have included detailed references and methods to produce our cell free protein synthesis system in the materials and methods section, detailed below.

“Cell extract preparation for cell-free protein synthesis

E. coli Origami™ B(DE3) (Novagen, 70837) extracts were prepared using a modified version of established protocols^{1,2}. Briefly, a 150 mL Origami™ B(DE3) starter culture was inoculated in LB from a glycerol stock and cultured in a 250 mL baffled flask at 37 °C for 16 hours. The 2xYTP was prepared without glucose in 75% of the final volume and sterilized using an autoclave. A 4x glucose solution was prepared and autoclaved separately, then added to the medium immediately before use. The starter cultures were used to inoculate 1 L of 2xYTPG media (16 g/L tryptone, 10 g/L yeast extract, 5 g/L sodium chloride, 7 g/L potassium phosphate dibasic, 3 g/L potassium phosphate monobasic, 18 g/L glucose) in a 2.5 L Full-Baffle Tunair shake flask at an initial OD600 of 0.08. Cells were cultured at 37 °C at 220 RPM in a shaking incubator. Cultures were grown until OD600 0.4-0.6, at which point the expression of T7 RNA polymerase was induced by the addition of IPTG to a final concentration of 0.5 mM. Cells were harvested at an OD600 of 2.5 via centrifugation at 12,000g for 1 minute at 4 °C. Cell pellets were washed three times with 25 mL S30 buffer per 50 mL culture (10 mM Tris Acetate pH 8.2, 14 mM Magnesium Acetate, and 60 mM Potassium Acetate). Pellets were resuspended in 1 mL S30 buffer per gram of cell mass. Cell suspensions were lysed using a single pass on an Avestin EmulsiFlex-B15 Homogenizer at a lysis pressure of 24,000 PSI. Cell debris was separated via centrifugation at 18,000g for 20 minutes, and the clarified lysate was collected, flash-frozen in liquid nitrogen, and stored at -80 °C.

Cell-free protein synthesis reactions

CFPS reactions were composed of the following reagents: 8 mM magnesium glutamate, 10 mM ammonium glutamate, 130 mM potassium glutamate, 1.2 mM ATP, 0.5 mM of

each CTP, GTP, and UTP. 0.03 mg/mL folinic acid, 0.17 mg/mL *E. coli* MRE600 tRNA (Roche 10109541001), 100 mM NAD, 50 mM CoA, 4 mM oxalic acid, 1 mM putrescine, 1 mM spermidine, 57 mM HEPES pH 7.2, 2 mM of each amino acid, 33.3 mM PEP, 20% v/v *E. coli* extract, varying concentrations of DNA template, and the remainder water. The preparation of these reagents has been described in detail elsewhere³. For DNA templates, plasmids were used at a concentration of 8 nM, and unpurified linear PCR products were used at 6.66 % (v/v). For the expression of antibodies, each template was added to a final concentration of 6.66 % (v/v). For antibody and sdFab expression 4 mM oxidized glutathione, 1 mM reduced glutathione, 14 μ M of purified DsbC, and 50 μ M FkpA were also supplemented to the reactions. Additionally, for oxidizing CFPS reactions, cell-extracts were treated with 500 μ M iodoacetamide (IAM) at room temperature for 30 minutes before use in CFPS⁴. All reaction components were assembled on ice and were either run as 12 μ L reactions in 1.5 mL microtubes or 2 μ L reactions in 384 well plates (BioRad, HSP3801). For 2 μ L reactions, components were transferred to the plate using an Echo 525 acoustic liquid handler. A mix containing all the CFPS components except for the DNA was dispensed from 384PP Plus plates (Labcyte, PPL-0200) using the BP setting. The DNA (unpurified PCR products) was dispensed from a 384LDV Plus plate (Labcyte, LPL-0200) using the GP setting. Reactions were allowed to proceed at 30 °C for 20 hours.”

3. Supplementary Figure 1 line 38 – describe GFP quantitation.

Thank you for pointing out this oversight. We have now described the GFP quantification approach in the methods. Specifically, we write:

“Quantification of cell-free protein synthesis reactions

To quantify sfGFP fluorescence, a standard curve was prepared using previously reported methods². Radioactive leucine was added to CFPS at a final concentration of 10 μ M of L-[¹⁴C(U)]-leucine (Perkin Elmer NEC279E250UC, 11.1GBq/mMole), followed by precipitation of the expressed proteins and scintillation counting⁵. To quantify sfGFP fluorescence, 2 μ L of a CFPS reaction was diluted in 48 μ L of water in a Black Costar 96 Well Half Area Plate. Fluorescence was measured using a BioTek Synergy™ H1 plate reader with excitation and emission wavelengths of 485 and 528, respectively. Scintillation counts and fluorescence were fit to determine a standard curve for use with non-radioactive samples.”

4. Supplementary Figure 2a – minus DTT – quantitate the % of fully intact disulphide linked IgG

We have modified the language in the manuscript to clarify that there are incomplete IgG assembly products. We have also included the % fully intact IgG in the manuscript. Specifically, we now write:

In the results section:

“Under the tested conditions, we observed additional bands corresponding to incomplete assembly of the IgG consistent with other reports that efficient assembly of full-length

antibodies in CFPS requires further optimization (e.g., temperature, DNA template ratio, and DNA template expression timing)⁶⁻¹¹.”

In the figure caption for Supplementary Figure 2:

“The full-length IgG band represents 14% of total antibody product measured by background subtracted densitometry.”

4 chain species. There appears to be some heterogeneity. Line 120 in main text (assembly of full IgG) is not supported by the SDS-PAGE result

We agree heterogeneity is present in the assembled product, but assembly of full-length IgG is supported through the observation of a full length heterotetrameric IgG band via SDS PAGE. We now highlight this band with a red arrow on the gel. Previous reports have highlighted the importance of optimizing cell-free protein synthesis reaction conditions to maximize protein yields, establishing an oxidizing environment to enable disulfide bond formation, expressing appropriate ratios of the heavy and light chains, and using temporal addition of heavy chain and light chain plasmids to aid full-length heterotetrameric IgG production. We have included additional text (already included above in response to the previous comment) to highlight this fact.

However, given the focus of our manuscript as a screen to evaluate antibody binding, we chose not to further optimize full length heterotetrameric IgG assembly. Our goal was to develop a method to enable high-throughput profiling of individual antibody binding regions, and for this reason, we focused on the sdFab format to maximize the probability of obtaining an intact binding domain for functional screening. To clarify this point, we write:

“Consistent assembly under the same experimental conditions is important for high-throughput screening since individual conditions cannot be optimized for hundreds or thousands of antibodies. We thus opted to use the synthetically dimerized antigen-binding fragment (sdFab, also called ecobodies^{6,8} or zipbodies⁸) format and, like Ojima-Kato *et al.*⁶⁻⁸, found that the assembly of sdFabs were more uniform than their corresponding Fabs in CFPS for a panel of antibodies (Supplementary Fig 2b-c).”

5. Supplementary Figure 2b/c – was this performed plus or minus DTT?. Different antibodies showed different levels of oxidised product – why?

We thank the reviewer for raising this point. All gels reported in this figure were run without DTT. This is now noted in the figure legend.

In terms of different levels of oxidized product, this is an excellent question, and one that is of central focus when manufacturing antibodies using CFPS. Different antibodies could show different levels of oxidized product for several reasons. First, we did not perform individual optimizations for each antibody to balance the expression of heavy and light chains. Thus, imbalanced expression of the two chains can lead to varied levels of oxidized product. Second, chains from different antibodies may fold more or less readily

in the CFPS environment, leading to differences in levels of assembled and oxidized product.

Given our goal of high-throughput screening, it was not possible to individually optimize expression levels for both chains for every antibody to achieve complete assembly of the chains. This is an additional reason for use of the sdFab format. Importantly, we observe sufficient assembled antibody to evaluate the functional binding for most of the antibodies screened in this manuscript.

6. Line 127 – “more consistent expression of zip-bodies” is not supported by Supplementary Figure 2c where there is some heterogeneity in intact product.

We thank the reviewer for raising this point but think there is confusion over the use of “assembly” versus “expression.” In the referenced text in our initial submission, we stated:

“Like reports by Ojima-Kato *et al.*¹⁷⁻¹⁹, we found that the assembly of synthetically dimerized antigen-binding fragments (sdFab, also called ecobodies^{17,19} or zipbodies¹⁸) were more consistent than their corresponding standard Fabs in CFPS for a small panel of antibodies and opted to utilize the sdFab format for expression (Supplementary Fig 2b-c).”

Here the assembly of sdFabs is more consistent in CFPS than of the corresponding Fabs, not the expression. Comparing the gels in Supplementary Figure 2b and 2c, there are a greater number of antibodies that show bands at the expected size corresponding to assembled dimer in the sdFab format than the Fab format. The reviewer is correct that heterogeneity (i.e., unassembled heavy and light chains) is still present in the sdFab format. However, given the discussion above about the goals of the screening platform, we believe that the claim that sdFabs assemble more consistently in CFPS is justified.

7. Supplementary Figure 3e – why does the Regeneron antibody show a low signal although the expression level looked OK in Supp. Fig 2c?

This is an excellent question. The AlphaLISA-based assembly metric requires the binding of an antibody to the constant domain of the human light chain of the expressed sdFab. It is possible that if there is misfolding of this domain, it could interfere with the assembly measurement. This would not be discernable via the SDS PAGE result in Supplementary Fig. 2c. A much deeper investigation into the structure of CFPS-derived REGN10933 antibody specifically would be required to determine this, which we believe is beyond the scope of our manuscript. In our updated manuscript, this result was reproduced in our repeated data, presented in Supplementary Fig. 8.

8. Supplementary Figure 4 requires a greater explanation as to what is meant by the y-axis “number of constructs” and the aim of this experiment.

Thank you for highlighting this point that requires greater explanation. This figure is intended to quantitatively describe the variability present in our workflow. Additionally, we

have added the following text in the figure caption to further clarify the meaning of the “number of constructs”:

“The number of constructs is the number of unique antibody fragments in a given histogram bin.”

Reviewer #2 (Remarks to the Author):

In this article, Hunt et al. report on the use of automated CFPS to enhance the throughput of antibody discovery. The ability of cell-free synthesis that can produce recombinant proteins without involving cell cultivation steps is truly useful for streamlining the discovery and engineering of novel proteins. In particular, the availability of high-throughput PPI assay methods, including the Alpha Screen used in this study, makes it more feasible to build a completely “cell-free” process of antibody discovery by implementing automated CFPS. Authors successfully automated the steps of cell-free expression and analysis of anti-SARS-CoV-2 antibodies. Presented results support the possibility of using the proposed approach to cope with emerging pandemic, which will draw attention of the broad scientific community.

We thank the reviewer for pointing out the utility and interest of our developed antibody screening workflow.

However, I would like to see several points clarified before its publication in Nature Communications.

1. Difficulties in proper folding and assembly of Fab have been a major issue in cell-free synthesis of antibodies. Authors utilized AlphaLISA to monitor antibody expression and assembly in CFPS, which is bit confusing. In their experiment, AlphaLISA for Fab assembly was designed to produce signal when the Fab chains on the donor bead and the acceptor bead are assembled to bring the two different beads in close proximity. How can it be assured that the assembly took place only between the Fab chains on different beads? Assembly of Fab chains on the same bead would not produce the AlphaLISA signal, and thus would lead to wrong evaluation of the targeted antibodies.

Thank you for raising this point for clarification. The antibodies are expressed in CFPS before addition of the AlphaLISA beads, so there is no influence of the AlphaLISA measurement on the assembly process itself. In the Assembly measurement, the light chain is linked to a donor bead through an anti-human light chain constant domain antibody and the heavy chain is linked to an acceptor bead via the C-terminal strepII or super FLAG tag. We have observed no cross-reactivity between these two immobilization strategies, and thus there is no opportunity for heavy chains to be immobilized on the donor bead or light chains to be immobilized on the acceptor bead. Thus, signal is only observed when heavy and light chain are associated. To highlight this point, we have included in the revised manuscript an additional negative control in Supplementary Fig. 5:

Supplementary Figure 5 | AlphaLISA for measuring assembly of CFPS derived sdFab. **a**, Assembly AlphaLISA measurement of a model anti-HER2 antibody fragment. Heavy and light chain were expressed either separately or together in a CFPS reaction. Only when both chains are co-expressed is assembly AlphaLISA signal observed. AlphaLISA signal is indicative of sdFab assembly, though the signal is subject to the “hook effect”¹², which can cause lower signal at higher concentrations. **b**, Assembly AlphaLISA measurement of firefly luciferase (fluc) as a non-antibody negative control and a panel of sdFabs.

2. The size of antibody library is not large enough to demonstrate the impact of the proposed approach. In fact, the set of 120 antibodies can be readily expressed and analyzed by conventional manual CFPS reactions. It would be desirable to expand the size of clonal library to justify the implementation of automated procedures.

We agree with the reviewer that 120 antibodies can be readily screened by conventional manual screening, either via CFPS or via other methods. However, such an approach would not inherently be easy to automation. We therefore sought to develop methods that were amenable to automated procedures, noting that the platform is not yet fully automated. In the revised manuscript, we have changed how automation is discussed, including removing the word automated from the title and including discussion of this point in the Discussion section, copied below.

“Second, all workflow steps were developed with automation and throughput in mind. Each step in the process consists of straightforward liquid handling and temperature incubation steps. Although the workflow as reported here is not fully automated and requires manual intervention to move plates between liquid handling steps, end-to-end automation using a robotic arm and incubators will be straightforward due to the workflow’s simplicity. Of note, the tools leveraged in this work are those that were developed for high-throughput small molecule drug discovery, a field where it is commonplace to evaluate 10^4 to 10^5 compounds¹³. Similar throughput should be attainable for high-throughput antibody screening using our reported methods with additional investment in automation.”

With regards to the number of antibodies screened, we agree it is desirable to expand the library of clonal sequences tested. Instead of screening large numbers of antibodies using a method that is not itself fully automated, we chose to highlight the speed and ease of the workflow by having a single researcher express and evaluate many antibodies in a short period of time (faster than would be possible in conventional cell-based methods). In our revised manuscript we have collaborated with members of the Diamond laboratory at Washington University (now co-authors) to screen an additional 119 novel antibodies identified by sequencing B cells derived from a mouse immunized with the SARS-CoV-2 spike protein. We have also included a set of high interest antibodies that contains all 8 historical emergency use authorized COVID-19 antibodies and 11 previously reported broadly neutralizing antibodies. Taken together, this more than doubles the number of antibodies evaluated in this work. These data are summarized in the new sections “Expression and evaluation of high interest SARS-CoV-2 antibodies” and “Discovery of SARS-CoV-2 antibodies from immunized mice” in the results section and their corresponding data are included below.

Fig. 3 | Evaluating high interest COVID-19 neutralizing antibodies. **a**, Heatmaps of the binding of EUA and bnAb sdFabs measured using AlphaLISA to detect S6P binding (\log_{10} scaled), RBD binding (\log_{10} scaled), and ACE2 competition (linearly scaled). **b**, Heatmap profiling binding of EUA and bnAb sdFabs against SARS-CoV-2 historical VOC and the coronaviruses SARS-CoV, MERS-CoV, HCoV-HKU1, HCoV-OC43, HCoV-NL63, HCoV-229E. Data are \log_{10} scaled. **a-b**, AlphaLISA data are the mean of 3 independent replicates ($n=3$). Antigen concentrations for AlphaLISA experiments listed in Supplementary Table 2.

Fig. 4 | Discovery of murine antibodies targeting SARS-CoV-2 using cell-free antibody screening. **a**, Summary of the S6P binding, RBD binding, ACE2 competition, and assembly AlphaLISA data for all 119 antibody fragments screened. **b**, Comparison of RBD AlphaLISA data and S6P AlphaLISA data. sdFabs with significant binding to either RBD or S6P are shown in black. **a-b**, Black data points represent data that are statistically significant (two-sided t-test FDR corrected $p < 0.05$) from background and have an average value > 3 standard deviations away from background. Grey data were considered not significantly different from background. **c**, Heatmaps of select antibody fragments detailing individual S6P binding, RBD binding, and ACE2 competition AlphaLISA data. S6P binding and RBD binding data are \log_{10} scaled and ACE2 competition is linearly scaled. **d**, ACE2 competition measured by titrating the concentration of ACE2. sdFabs in crude CFPS were diluted to a fixed concentration and combined with a fixed concentration of RBD and varying concentrations of ACE2. **e**, ACE2 bridging by an sdFab and the RBD. A fixed concentration of ACE2 and the RBD was combined with varying concentrations of sdFab. **d-f** Error bars represent the standard error of the mean. An absence of error bars indicates error within the marker. **f**, Heatmap profiling select antibodies for binding against SARS-CoV-2 historical VOC and SARS-CoV. Data are \log_{10} scaled. **a-f**, AlphaLISA data are the mean of 3 independent replicates ($n=3$). Antigen concentrations for AlphaLISA experiments listed in Supplementary Table 2.

3. Are there specific reasons that authors used different liquid handling equipment for Gibson assembly and CFPS? Especially when considering the number of antibodies tested, I am not sure about the advantage of combining conventional liquid handling robot and the Echo system.

This is a good question and there is indeed a reason relating to liquid volume. The Echo is an excellent tool for dispensing small volumes and combinatorally mixing components. As such, we find it to be valuable for workflow development as well as reducing reagent consumption. However, the Echo cannot perform transfers larger than several microliters. These types of transfers are invaluable for other parts of the protocol (e.g., setting up Echo source plates, diluting Gibson assembly reactions prior to PCR (20-fold dilution), diluting CFPS prior to AlphaLISA (10-fold dilution). This is more easily accomplished with a traditional tip-based liquid handler like the Integra ViaFlo used for the non-Echo transfer steps.

4. While authors performed their work with a collection of cloned genes, it will make their claims more convincing if they can demonstrate the proposed workflow can be adopted for the discovery of antibodies starting from B cells, which will require the integration of single cell RT-PCR step.

Thank you for this suggestion. We agree that the integration of RT-PCR to recover B cell heavy and light chains would be a valuable addition to this workflow. However, the step of B cell RT-PCR in combination of expression of an antibody in CFPS has already been demonstrated in multiple other contexts^{17,19,41}, which we have noted in the Discussion of the revised manuscript:

“Additionally, we utilized synthetic DNA coding for the antibodies of interest as opposed to PCR products from single B cells coding for the paired heavy and light chains. However, previous work suggests that this workflow is compatible with PCR products amplified from single B cells from an immunized animal^{6,8,14}.”

We do not believe that replicating these findings adds appreciably to the present work, as our focus was on eliminating bottlenecks in the workflow to express and characterize binders. Additionally, given the falling costs of DNA synthesis and faster turnaround times (e.g., IDT now offers eblocks for 6 cents/base with a 2–3-day turnaround time) we believe that DNA synthesis is an increasingly viable option when paired with sequencing of a sorted B cell repertoire¹⁵. This is the approach we used to screen antibodies derived from an immunized mouse in Figure 4.

Reviewer #3 (Remarks to the Author):

In this manuscript, the authors describe an integrated pipeline for antibody expression and characterization, including four key developments. The high-throughput and cell-free antibody screening workflow was validated using 120 published antibodies.

Major comments:

1. This study used “previously published antibodies” to show the platform is functional. However, the authors should apply the platform in a discovery screen and demonstrate that it can be used to rapidly identify “novel” neutralizing antibodies against SARS-CoV-2 or other pathogens from B cells of convalescent patients within several hours.

We appreciate the interest of the reviewer for showing that the platform can be used to identify novel antibodies. To address this comment, we collaborated with members of the Diamond laboratory at Washington University (now co-authors) to screen anti-SARS-CoV-2 antibodies isolated from mice immunized with the SARS-CoV-2 spike protein. We screened 119 previously unreported mouse antibodies and identified several candidate neutralizing antibodies. The details of this screen are presented in a new Results section in the revised manuscript and included below. Like the previous work in the manuscript, the antibodies were expressed and evaluated in less than 24 hours.

“Discovery of SARS-CoV-2 antibodies from immunized mice

We next sought to discover previously unidentified antibodies against SARS-CoV-2 using our workflow. We immunized mice with the Wu-Hu-1 SARS-CoV-2 S6P and isolated memory B cells using fluorescence activated cell sorting. The pooled sorted B cells were sequenced to identify paired heavy and light chains, which were codon optimized and ordered commercially as synthetic DNA. We screened 119 identified antibody sequences measuring S6P binding, RBD binding, ACE2 competition, and sdFab assembly (Fig. 4). Like our initial screen with previously reported antibodies, a high consistency between independent experimental replicates was observed (Supplementary Fig. 11) with average coefficients of variation between 0.09 and 0.22 depending on the AlphaLISA measurement modality.

We observed assembly signal for all 119 antibody fragments and S6P or RBD binding for 37 of the screened antibody fragments (Fig. 4a), with some antibody fragments only exhibiting binding to either the RBD or S6P (Fig. 4b). Several of the antibody fragments competed with ACE2 in the initial screen (Fig. 4a, 4c and Supplementary Fig. 12), which we further validated via dose response analysis (Fig. 4d). SC2-9 exhibited strong competition with ACE2 (required high concentrations of ACE2 for inhibition of binding to the RBD) and is likely the highest affinity potential neutralizing antibody against Wu-Hu-1 SARS-CoV-2 identified in our screen. The antibody SC2-3 required high concentrations of ACE2 to inhibit binding to the RBD and exhibited RBD bridging with ACE2 (Fig. 4e), like that observed for S309 (Supplementary Fig. 9). Combined, these data suggest that SC2-3 either only weakly competes with ACE2 or that binding of SC2-3 to its target epitope may influence ACE2 binding. In the ACE2 bridging experiment, the reduction in AlphaLISA signal at higher concentrations likely is due to the “hook effect”¹² where binding sites on the AlphaLISA beads become saturated, and higher concentrations of antibody begin to inhibit signal. We also profiled the binding of these antibodies to historical SARS-CoV-2 VOC and SARS-CoV spike proteins (Fig. 4f). Most tested antibodies exhibited reduced binding to several different VOC spike proteins, except for SC2-3, which

exhibited binding to all tested SARS-CoV-2 VOC spike proteins. These data indicate that our developed workflow can be used to discover antibodies that bind a target antigen.”

Fig. 4 | Discovery of murine antibodies targeting SARS-CoV-2 using cell-free antibody screening. **a**, Summary of the S6P binding, RBD binding, ACE2 competition, and assembly AlphaLISA data for all 119 antibody fragments screened. **b**, Comparison of RBD AlphaLISA data and S6P AlphaLISA data. sdFabs with significant binding to either RBD or S6P are shown in black. **a-b**, Black data points represent data that are statistically significant (two-sided t-test FDR corrected $p < 0.05$) from background and have an average value > 3 standard deviations away from background. Grey data were considered not significantly different from background. **c**, Heatmaps of select antibody fragments detailing individual S6P binding, RBD binding, and ACE2 competition AlphaLISA data. S6P binding and RBD binding data are \log_{10} scaled and ACE2 competition is linearly scaled. **d**, ACE2 competition measured by titrating the concentration of ACE2. sdFabs in crude CFPS were diluted to a fixed concentration and combined with a fixed concentration of RBD and varying concentrations of ACE2. **e**, ACE2 bridging by an sdFab and the RBD. A fixed concentration of ACE2 and the RBD was combined with varying concentrations of sdFab. **d-f** Error bars represent the standard error of the mean. An absence of error bars indicates error within the marker. **f**, Heatmap profiling select antibodies for binding against SARS-CoV-2 historical VOC and SARS-CoV. Data are \log_{10} scaled. **a-f**, AlphaLISA data are the mean of 3 independent replicates

(n=3). Antigen concentrations for AlphaLISA experiments listed in Supplementary Table 2.

General comments:

1. As the authors mention, this workflow has advantages of higher speed and throughput when performing discovery screens for antibodies from single B cells. Can this platform also be adapted for use with hybridoma technologies or phage display libraries?

We share the Reviewer's interest in extending the utility of the workflow beyond antibodies derived from B cells. We anticipate the workflow can be adapted to both different binding protein formats and antibody discovery methodologies. Our workflow simply requires DNA coding for the protein of interest with the appropriate DNA adaptors for Gibson assembly into our vector. Our strategy could, for example, easily be applied to phage display or hybridomas either by sequencing the selected candidates and ordering the DNA (as we have done for the murine antibodies derived from B cells included in the revised manuscript) or by amplifying the VH/VL domains of the selected antibodies via PCR and using our DNA assembly method. We have added additional discussion in the updated manuscript to address these points.

"Additionally, we utilized synthetic DNA coding for the antibodies of interest as opposed to PCR products from single B cells coding for the paired heavy and light chains. However, previous work suggests that this workflow is compatible with PCR products amplified from single B cells from an immunized animal^{6,8,14}. Similarly, the workflow is agnostic to the method used for antibody selection, and thus is likely also compatible with other technologies beyond the isolation of single B cells like hybridomas and *in vitro* display techniques¹⁶. We also recently applied similar methods to those described here to identify highly potent computationally designed minibinders that neutralize SARS-CoV-2¹⁷, which suggests that this approach may be broadly applicable to different binding protein formats."

2. The authors show a correlation between assembly, protein-protein interactions and competition. For cases such as REGN10987, which had low Alpha signal of assembly and RBD binding, would the antibody be selected in an actual high-throughput screen?

As we have outlined above, we view this platform as a tool to improve the success rate of antibodies that are evaluated using more labor-intensive processes. Additionally, given the strong preference we have observed using the selected screening conditions towards high affinity antibodies, signal above background likely indicates a promising hit. Taking the Brouwer et al. data set as an example, there are relatively few high affinity hits in this data set. In a hypothetical high-throughput screen of the Brouwer et al. data set with REGN10987 included, we would select REGN10987 for further evaluation.

In our revised manuscript, we have repeated the experiments involving the historically emergency use authorized antibodies and included all 8 antibodies in this category. In the process, we performed several additional optimizations to the screening workflow including adjusting the tag used for immobilization of the sdFab in the AlphaLISA

experiment (changed from strepII to sFLAG) as well as the concentrations of antigens used in the AlphaLISA experiments. We found that increasing the concentration of the target antigen as well as utilizing biotinylated antigens yielded more robust results, including for the Regeneron antibodies. We adopted these conditions for Figures 3 and 4.

3. An AlphaLISA method is used to characterize protein-protein interactions and neutralizing antibodies. The authors stated that decreased AlphaLISA signals indicate interference with protein-protein interactions; however, in Supplementary Figure 3c, the higher concentration of “cell lysate” sample had a higher signal. Do increases in signal have some meaning such as improvement of protein-protein interaction?

The AlphaLISA measurement and PPIs are sensitive to the biophysical environment and thus signal can change because of changing factors like salts, crowding agents, etc. All high-throughput screening AlphaLISA assays were performed under the same biophysical conditions (concentration of buffer, CFPS, etc.) and thus are comparable across samples.

The increase in signal due to the presence of the cell lysate is a particularly interesting phenomenon, as it is only observed in the condition where an anti-his acceptor bead is used (Supplementary Figure 3c) and not when the nickel chelate acceptor bead is used (Supplementary Figure 3b). Anecdotally we have observed that an anti-FLAG acceptor bead is not impacted by the cell lysate. One hypothesis we have is that this is related to increased crowding effects due to the protein content of the cell lysate (which contains around 30 mg/mL cytoplasmic *E. coli* protein), which could influence the protein-protein interaction. However, it is unclear how this would specifically impact the anti-his interaction.

Supplementary Figure 4 | The impact of CFPS reagents on AlphaLISA signal. a-c, Evaluation of the effect of CFPS reagents on AlphaLISA. Concentrations are plotted as v/v fraction of the final concentration of the reagent in a CFPS reaction. Reagents were diluted in water at the concentration they normally reside at in CFPS. Reagents were tested in mixtures that were used to assemble CFPS reactions. The salt solution contains

8 mM magnesium glutamate, 10 mM ammonium glutamate, and 130 mM potassium glutamate. Master Mix contains 1.2 mM ATP, 0.85 mM GTP, 0.85 mM UTP, 0.85 mM CTP, 0.03 mg/mL folinic acid, and 0.17 mg/ml *E. coli* tRNA. Reagent Mix contains 0.4 mM NAD, 0.27 mM CoA, 4 mM oxalic acid, 1 mM putrescine, 1.5 mM spermidine, and 57 mM HEPES. Amino Acid Mix contains 2 mM of all 20 amino acids. PEP is 30 μ M phosphoenolpyruvate. LET is 0.066 v/v fraction unpurified PCR mix containing the LET for sfGFP. **a**, Evaluation of the effect of CFPS reagents on AlphaLISA detection chemistry using the TrueHits kit. Biotin and Streptavidin labeled beads associate directly with one another and serve as a control for reagents impacting the AlphaLISA measurement chemistry. **b**, Evaluation of the effect of CFPS reagents on AlphaLISA detection of the SARS-CoV-2 RBD and ACE2 interaction measured by the Protein A donor bead and Ni Chelate acceptor bead. **c**, Evaluation of the effect of CFPS reagents on AlphaLISA detection of the SARS-CoV-2 RBD and ACE2 interaction measured by the Protein A donor bead and anti-6xhis acceptor bead.

4. Does protein aggregation affect the signal of the AlphaLISA (leading to false positives)? Did aggregation occur in this study? How do the authors rule out this possibility?

Aggregation could theoretically affect the AlphaLISA signal, but it would be unlikely. To do so, both the binding protein (e.g., sdFab) and the target (e.g., RBD, SP6) must associate into an aggregate structure and maintain accessible tags for immobilization and not preclude association with the donor and acceptor beads.

In the binding and competition experiments, false positives should not occur because protein expression of the sdFab (synthesis folding, and dimerization) is carried out separately from the introduction of a purified soluble binding target (e.g., RBD, SP6).

In the assembly experiments, it is possible that aggregation of the heavy and light chains could cause a false positive signal (again both fragments would have to aggregate together while their peptide tags remain fully accessible for binding to the beads). Although we cannot rule out aggregation in the assembly test, SDS PAGE analysis (Supplementary Figure 2c) indicates dimerized sdFab as a major product on a native gel.

5. In Figure 1b-d, the authors mentioned that they tested five different commercial antibodies (Line 150 of page 8), but in Figure 1c and d, there are only four antibodies shown.

Thank you for pointing out this error. We have corrected this in the text.

6. Statistical tests are needed for Figure 2c-f.

We thank the reviewer for this suggestion and have computed the Pearson correlation coefficient of the comparisons we have drawn with the Brouwer *et al.* data set. We have included the following text analyzing these data.

“While there is only a weak correlation between our AlphaLISA data and the corresponding Brouwer *et al.* S6P binding ELISA data, the RBD binding ELISA data, and the pseudovirus neutralization data (Pearson correlation coefficients with 95% confidence interval and p value of $r = 0.35$ (0.15 to 0.53, $p = 0.01$), $r = 0.41$ (0.22 to 0.58, $p < 0.0001$), and $r = 0.40$ (0.20 to 0.56, $p = 0.002$) for Fig. 2c, 2d, and 2e respectively), the screening conditions used consistently identified the strongest binders and most potent neutralizing antibodies. Collectively, these data show that our workflow can be used to express and evaluate human antibody fragments as a filter to select potential candidates for further development.”

7. “The Salt Solution and Amino Acids Mix inhibit immobilization of his-tagged proteins on the NiChelate bead.” in legend of Supplementary Figure 3c is redundant.

Thank you for the suggestion, the text has been removed in the figure caption.

8. The authors need to describe how they determined the concentration and percentage of assembled sdFab in the AlphaLISA system.

The concentration and percentage of assembled sdFab was not determined using the AlphaLISA system. Our intention with the sdFab assembly AlphaLISA measurement was to provide a qualitative metric to help observe sdFabs that expressed poorly or did not assemble. To clarify the intent and usage of the assembly AlphaLISA measurement, we have added the following to the text to the results section describing the development of this assay:

“While the main assay focus was on protein binding and competition, we utilized the assembly assay as a qualitative positive control to confirm that antibodies expressed and assembled in the CFPS reaction.”

Response to Reviewers References

1. Kwon, Y. C. & Jewett, M. C. High-throughput preparation methods of crude extract for robust cell-free protein synthesis. *Sci. Rep.* **5**, 1–8 (2015).
2. Chen, Z. *et al.* De novo design of protein logic gates. *Science* **368**, 78–84 (2020).
3. Silverman, A. D., Kelley-Loughnane, N., Lucks, J. B. & Jewett, M. C. Deconstructing Cell-Free Extract Preparation for in Vitro Activation of Transcriptional Genetic Circuitry. *ACS Synth. Biol.* **8**, 403–414 (2019).
4. Matsuda, T., Watanabe, S. & Kigawa, T. Cell-free synthesis system suitable for disulfide-containing proteins. *Biochem. Biophys. Res. Commun.* **431**, 296–301 (2013).
5. Swartz, J. R., Jewett, M. C. & Woodrow, K. A. Cell-Free Protein Synthesis With Prokaryotic Combined Transcription-Translation. in *Recombinant Gene Expression: Reviews and Protocols* (eds. Balbás, P. & Lorence, A.) 169–182 (Humana Press, 2004). doi:10.1385/1-59259-774-2:169.
6. Ojima-Kato, T., Nagai, S. & Nakano, H. Ecobody technology: rapid monoclonal antibody screening method from single B cells using cell-free protein synthesis for antigen-binding fragment formation. *Sci. Rep.* **7**, 13979 (2017).
7. Ojima-Kato, T. *et al.* ‘Zipbody’ leucine zipper-fused Fab in *E. coli* in vitro and in vivo expression systems. *Protein Eng. Des. Sel.* **29**, 149–157 (2016).
8. Ojima-Kato, T. *et al.* Rapid Generation of Monoclonal Antibodies from Single B Cells by Ecobody Technology. *Antibodies* **7**, 38 (2018).
9. Murakami, S., Matsumoto, R. & Kanamori, T. Constructive approach for synthesis of a functional IgG using a reconstituted cell-free protein synthesis system. *Sci. Rep.* **9**, 671 (2019).
10. Martin, R. W. *et al.* Development of a CHO-Based Cell-Free Platform for Synthesis of Active Monoclonal Antibodies. *ACS Synth. Biol.* **6**, 1370–1379 (2017).
11. Groff, D. *et al.* Engineering toward a bacterial “endoplasmic reticulum” for the rapid expression of immunoglobulin proteins. *MAbs* **6**, 671–678 (2014).
12. Newton, P., Harrison, P. & Clulow, S. A novel method for determination of the affinity of protein: protein interactions in homogeneous assays. *J. Biomol. Screen.* **13**, 674–682 (2008).
13. Schorpp, K. *et al.* Identification of small-molecule frequent hitters from AlphaScreen high-throughput screens. *J. Biomol. Screen.* **19**, 715–726 (2014).
14. Ding, R. *et al.* Rapid isolation of antigen-specific B-cells using droplet microfluidics. *RSC Adv.* **10**, 27006–27013 (2020).
15. Tanno, H. *et al.* A facile technology for the high-throughput sequencing of the paired VH:VL and TCR β :TCR α repertoires. *Sci. Adv.* **6**, eaay9093 (2020).
16. Laustsen, A. H., Greiff, V., Karatt-Vellatt, A., Muyldermans, S. & Jenkins, T. P. Animal Immunization, in Vitro Display Technologies, and Machine Learning for Antibody Discovery. *Trends Biotechnol.* **xx**, 1–11 (2021).
17. Hunt, A. C. *et al.* Multivalent designed proteins neutralize SARS-CoV-2 variants of concern and confer protection against infection in mice. *Sci. Transl. Med.* eabn1252 (2022) doi:10.1126/scitranslmed.abn1252.

REVIEWERS' COMMENTS

Reviewer #1 (Remarks to the Author):

The authors have addressed my points sufficiently for me to recommend that this report is published subject to minor corrections. Specifically, I would request that the answer to my question 7 "Supplementary Figure 3e – why does the Regeneron antibody show a low signal although the expression level looked OK in Supp. Fig 2c?" is addressed by additions to the manuscript text.

Reviewer #3 (Remarks to the Author):

The authors have answered most of the questions. However, the Major comments and most important questions raised by three reviewers did not responded. The authors should apply this platform in a large-scale antibody screen and demonstrate that it can be used to rapidly identify neutralizing antibodies against SARS-CoV-2 or other pathogens, but not only just used well-known or identified antibodies.

I expect that the number of antibody clones screened should be over 500, which would more accurately align with the "high-throughput" spirit of this article. This would also better reflect the real situation of number of antibody clones screened, whether using hybridoma or single B cell technologies. Additionally, this point echoes the comments by Reviewer 2, stating that synthesizing and screening 120 antibodies using traditional manual CFPS methods is not difficult.

REVIEWER COMMENTS

Reviewer #1 (Remarks to the Author):

The authors have addressed my points sufficiently for me to recommend that this report is published subject to minor corrections. Specifically, I would request that the answer to my question 7 "Supplementary Figure 3e – why does the Regeneron antibody show a low signal although the expression level looked OK in Supp. Fig 2c?" is addressed by additions to the manuscript text.

Thank you for this suggestion, we agree that including a discussion of this result adds valuable context to the assembly assay. We have included a discussion of this result when the assembly assay is introduced in the section "Integration of the AlphaLISA PPI assay to evaluate antibody fragment binding and assembly":

"Furthermore, we utilized AlphaLISA to develop a sdFab assembly screen to monitor antibody fragment expression and assembly in CFPS, a step that traditionally requires SDS-PAGE. The measurement immobilizes the heavy and light chains of the sdFab to the AlphaLISA beads, resulting in signal when the two chains are assembled (**Supplementary Fig. 5a**). The AlphaLISA assembly assay generally shows consistent prediction of sdFab assembly with SDS-PAGE on a panel of sdFabs and can thus be used to identify when sdFab expression or assembly fails (**Supplementary Fig. 5b**). In this panel, the sdFab REGN10933 yielded lower assembly signal than the other tested antibodies despite a strong band present by SDS-PAGE (**Supplementary Fig. 2c**). This could be a result of misfolding of the light chain constant domain, leading to reduced binding of the anti-light chain antibody to the REGN10933 light chain and thus lower assembly signal. However, deeper structural analysis of CFPS-derived REGN10933 would be required to understand this result further. Accordingly, we utilized the assembly assay as a qualitative positive control to confirm that antibodies expressed and assembled in the CFPS reaction and did not attempt to use the assay to quantify assembled antibody yields."

Reviewer #3 (Remarks to the Author):

The authors have answered most of the questions. However, the Major comments and most important questions raised by three reviewers did not responded. The authors should apply this platform in a large-scale antibody screen and demonstrate that it can be used to rapidly identify neutralizing antibodies against SARS-CoV-2 or other pathogens, but not only just used well-known or identified antibodies.

I expect that the number of antibody clones screened should be over 500, which would more accurately align with the "high-throughput" spirit of this article. This would also better reflect the real situation of number of antibody clones screened, whether using hybridoma or single B cell technologies. Additionally, this point echoes the comments by Reviewer 2, stating that synthesizing and screening 120 antibodies using traditional manual CFPS methods is not difficult.

Following guidance from the Editor, we have modified the language throughout our manuscript to remove the claim that the method is high-throughput. We have emphasized the possibility of the method to be high-throughput with further development of automation tools in the discussion.